# FROM LANGUAGE MODELING TO INSTRUCTION FOLLOWING: UNDERSTANDING THE BEHAVIOR SHIFT IN LLMS AFTER INSTRUCTION TUNING

## ABSTRACT

Large Language Models (LLMs) have achieved remarkable success, demonstrating powerful instruction-following capabilities across diverse tasks. Instruction tuning is critical in enabling LLMs to align with user intentions and effectively follow instructions. In this work, we investigate how the instruction tuning modifies pre-trained models, focusing on two perspectives: instruction recognition and knowledge evolution. To study the behavior shift of LLMs, we employ a suite of local and global explanation methods, including a gradient-based approach for input-output attribution and techniques for interpreting patterns and concepts in self-attention and feed-forward layers. Our findings reveal three significant impacts of instruction tuning: 1) It empowers LLMs to better recognize the instruction parts from user prompts, thereby facilitating high-quality response generation and addressing the "lost-in-the-middle" issue observed in pre-trained models; 2) It aligns the knowledge stored in feed-forward layers with user-oriented tasks, exhibiting minimal shifts across linguistic levels. 3) It facilitates the learning of word-word relations with instruction verbs through the self-attention mechanism, particularly in the lower and middle layers, indicating enhanced recognition of instruction words. These insights contribute to a deeper understanding of the behavior shifts in LLMs after instruction tuning and lay the groundwork for future research aimed at interpreting and optimizing LLMs for various applications[1].

## 1 INTRODUCTION

The remarkable capability of Large Language Models (LLMs) to align with user intentions is well-recognized across various real-world applications, where they are expected to be helpful, honest, and harmless AI assistants (Ouyang et al., 2022; OpenAI, 2023). Central to these roles, being "helpful" is the most fundamental requisite, emphasizing that LLMs should help users to complete various tasks, known as the "instruction following" capability. Many studies (Raffel et al., 2020; Wang et al., 2022; Zhou et al., 2023) show that *instruction tuning*, also called supervised fine-tuning (Ouyang et al., 2022), is critical to acquire such capability, by fine-tuning pre-trained models on high-quality prompt-response pairs. However, the impact of instruction tuning on the helpfulness of language models remains inadequately understood, limiting the improvements toward better-fine-tuned models.

While many studies on interpreting LLMs have delved into pre-trained models (Dai et al., 2021; Elhage et al., 2021; Olsson et al., 2022; Meng et al., 2022), in-context learning (Xie et al., 2021; Olsson et al., 2022; Liu et al., 2023), and single-task fine-tuning (Kokalj et al., 2021; Wu & Ong, 2021; Enguehard, 2023), their findings cannot be extrapolated to instruction tuning. This is because instruction tuning updates pre-trained weights (analysis of pre-trained weights and in-context learning paradigm does not hold.) to achieve robust generalization across a range of downstream tasks (analysis of single-task fine-tuned models does not hold). Therefore, we seek to explore how large language models undergo a shift in behavior—from primarily modeling language to effectively following instructions—after being subjected to the instruction tuning process.

In this work, we focus on answering the research question of how instruction tuning changes the pre-trained model's behavior from two perspectives, namely *instruction recognition* and *knowledge evolution*. In particular, what role do user instructions play when an instruction tuned model generates responses? Also, how does the encoded knowledge evolve after instruction tuning? We consider

---

[1]We will release our code and data upon acceptance.

these two perspectives under a well-accepted belief (Liang et al., 2023; Kung & Peng, 2023; Zhou et al., 2023) that an LLM becomes a helpful assistant by first recognizing human instructions and then retrieving corresponding knowledge to respond.

The primary way we study these questions is comparing the human-understandable explanations of pre-trained and fine-tuned models. However, it is still non-trivial to technically interpret recent large language models. For one, LLMs demonstrate their instruction following abilities by generating textual responses, yet many explanation methods for language models are designed for text classification (Sundararajan et al., 2017; Mu & Andreas, 2020; Wu & Ong, 2021; Enguehard, 2023). Moreover, instruction fine-tuned models retain massive knowledge from various topics and domains, but existing knowledge probing methods are limited to a specific task or domain (Press et al., 2019; Dai et al., 2021; Meng et al., 2022). To fill these gaps, we develop a series of local and global explanation methods as a *toolbox* to study large language models. Specifically, we first introduce a gradient-based local explanation method to determine the attribution of input tokens in the prompt to output tokens in the response. Next, we propose two global explanation methods to interpret the *textual patterns* and *concepts* encoded in self-attention as well as feed-forward layers with natural language. We further adopt a scalable automated interpretation method to interpret entire LLMs with billions of parameters. Our interpretation tools elucidate the crucial behavior shift in LLMs as they transition from language modeling to instruction following. While our emphasis is on behavior shifts after instruction tuning, future research might also apply our toolbox to understand LLMs for various other purposes.

To this end, we obtain three main findings related to instruction tuning as follows:

- **Finding-1:** *Instruction tuning enables models to recognize instruction words from user input prompts, guiding high-quality response generation.* We study the influence of input prompts by detailing their instruction and context words. Considering the input prompt "Fix grammar: me and him goes to an stores for buy a apples.", the instruction words are "Fix grammar:", while the rest is the context words. We introduce a gradient-based method to evaluate how much each input word contributes to output words, where we observe that key instruction words constantly guide the generation process, while the models perform a "match-and-reference" behavior on context words (Sec. 4.1). Based on this, we propose a density score to inherently assess if the model aligns with user intentions by computing the density of importance on instruction words (Sec. 4.2). Moreover, we explore the effects of prompt position by extending the density score to all input words and show that instruction tuning alleviates the "lost-in-the-middle" issue seen in pre-trained models (Sec. 4.3), indicating that the instruction tuned models could better utilize information presenting in the middle parts of input sentences compared with the pre-trained models.

- **Finding-2:** *Instruction tuning aligns the knowledge in feed-forward layers with user-oriented tasks, with minimal shifts across linguistic levels.* We study the knowledge in feed-forward layers at the "concept" level by interpreting the principal components of their stored patterns with human language (Sec. 5.1). Our analysis of these concepts spans two dimensions (Sec. 5.3): four common user-oriented tasks [2] where users need help from LLMs, and four linguistic levels [3] defined by the disciplines of the linguistic subject (Thomas, 2005). Since these linguistic levels were proposed to study natural language systematically, we introduce them as the control set to study whether LLMs' understanding of human language changes after instruction tuning. A notable shift is observed concerning user-oriented tasks, where the fine-tuned model more explicitly adjusts the concepts toward specific downstream tasks such as writing, coding, and solving math problems. In terms of linguistic levels, while there is no shift after instruction tuning, we observe an interesting phenomenon: as model depth increases, the proportion of semantic knowledge initially grows, but in the last few layers, it drops significantly while that of morphology knowledge (e.g., prefix "the-" and suffix "-ed") increases. This observation differs from the conventional belief that deeper layers mainly capture high-level concepts (Zeiler & Fergus, 2014; Selvaraju et al., 2016).

- **Finding-3:** *Instruction tuning encourages self-attention heads from lower layers to learn word-word relations with instruction verbs.* We study the knowledge in self-attentions by identifying word pairs that strongly activate corresponding column vectors in their query and key matrices, which pairs are also likely to co-occur within a context. (Sec. 6.1). We observe

---

[2]User-oriented tasks include "writing", "coding", "translation", and "solving math problem".
[3]Linguistic levels include "phonology", "morphology", "syntax", and "semantic".

that (Sec. 6.3) only approximately 65% of the word pairs from the same self-attention head remained consistent after instruction tuning. Furthermore, when analyzing the shifts in word pairs of instruction verbs (e.g., classify, summarize, translate) versus universal frequent verbs, we notice a higher percentage of self-attention heads from lower and middle layers stored instruction verb relations after fine-tuning, indicating that pre-trained models become adept at recognizing instruction words during instruction tuning.

This research not only provides empirical evidence demonstrating the significance of instruction words for human alignments, but also explains how self-attention and feed-forward networks distinctively contribute to this capability. In the following, we present notations and general experiment settings, followed by methods and results of each main finding, concluding with instruction tuning tips.

## 2 RELATED WORK

**Interpreting Language Models.** Interpreting LLMs has received significant attention in recent years. Majority investigations aimed to understand the decision-making processes of LLMs for a specific task or dataset, which involves feature attribution methods (Li et al., 2015; Vig, 2019; Kokalj et al., 2021), attention-based methods (Vig, 2019; Barkan et al., 2021), and sample-based methods (Kim et al., 2018; Wu et al., 2021). With the emergent capabilities of LLMs, a wave of research turned to understanding why LLMs could perform in-context learning (Xie et al., 2021; Olsson et al., 2022; Li et al., 2023; Wei et al., 2023; Xiong et al., 2023; Duan et al., 2023; Varshney et al., 2023), where they typically conduct empirical control experiments to prove their hypotheses. In parallel, some delved deeper into the core components of LLMs, including the self-attention mechanism (Elhage et al., 2021; Sukhbaatar et al., 2019) and feed-forward networks (Press et al., 2019; Geva et al., 2020; Voita et al., 2023; Petroni et al., 2019; Meng et al., 2022; Huang et al., 2023). Recent work also use LLMs as tools to enhance the interpretability of themselves (Bills et al., 2023; Singh et al., 2023). Our work builds on these foundations, introducing novel interpretation methods tailored for open-domain text generation in massive-size LLMs.

**Interpreting Instruction-tuned Models.** Recent studies in instruction tuning have uncovered various unexpected phenomena. A notable example is the "lost-in-the-middle" effect identified by Liu et al. (2023), which demonstrates that information presented in the middle of prompts often results in the poorest model performance, challenging the ability of instruction-tuned models to effectively utilize presented information. In contrast, Zhou et al. (2023) showed that even a limited set of prompt-response pairs could significantly enhance the instruction-following capabilities of large language models. Also, the practical parameter efficient training approach involving fine-tuning only the self-attention modules, such as LoRA (Juletx, 2023), highlights the distinct roles of self-attention versus feed-forward networks. Moreover, researchers (Liang et al., 2023; Kung & Peng, 2023; Zhou et al., 2023) have explored the impact of varying the complexity and format of instruction prompts during the instruction tuning of pre-trained models. Their findings reveal that models just learn superficial patterns through instruction tuning. These observations collectively motivate a deeper investigation into the internal dynamics of instruction-tuned models, aiming to reach a more comprehensive understanding that recognizes these diverse phenomena under a unified perspective.

## 3 PRELIMINARY

### 3.1 TRANSFORMER ARCHITECTURE

Considering $\mathcal{V}$ as a pre-defined vocabulary set, then $X$ denotes an $N$-length prompting text and $Y$ is a $M$-length response from a transformer-based language model $f$, where each individual token $x_n \in X$ or $y_m \in Y$ comes from $\mathcal{V}$. $f$ is defined in a $D$-dimensional space, starting with an input word embedding $\mathbf{E}_i \in \mathbb{R}^{|\mathcal{V}| \times D}$ presenting input tokens in $\mathbf{X} \in \mathbb{R}^{N \times D}$. $\mathbf{X}$ goes through $L$ transformer blocks, each containing a self-attention module and a feed-forward network. Every self-attention module includes $H$ heads that operate in a space with $D'$ dimensions. Each self-attention head captures word relations by $\mathbf{A}^h = \text{softmax}(\mathbf{X}\mathbf{W}_q^h(\mathbf{X}\mathbf{W}_k^h)^\top / \epsilon)$, where $\mathbf{W}_q^h, \mathbf{W}_k^h \in \mathbb{R}^{D \times D'}$ and $\epsilon$ is a constant. The aggregation of heads' outputs is $[\mathbf{A}^1 \mathbf{X}\mathbf{W}_v^1; ...; \mathbf{A}^H \mathbf{X}\mathbf{W}_v^H]\mathbf{W}_o$. Each feed-forward network is defined as $\sigma(\mathbf{X}\mathbf{W}_u^\top)\mathbf{W}_p$, where $\sigma$ refers to a non-linear function, and $\mathbf{W}_u, \mathbf{W}_p \in \mathbb{R}^{D'' \times D}$. At the end, the processed word embeddings dot product with the transpose of output word embedding matrix $\mathbf{E}_o \in \mathbb{R}^{|\mathcal{V}| \times D}$ for the next word prediction.

### 3.2 GENERAL EXPERIMENTAL SETTINGS

**Language Models.** We choose LLaMA family (Touvron et al., 2023) as the focus for two reasons. Firstly, LLaMA stands out as one of the most advanced publicly accessible pre-trained language model families. Secondly, LLaMA serves as the foundation for many instruction fine-tuned models, providing a vast array for further research. In this research, we mainly use the fully fine-tuned versions of Alpaca (Taori et al., 2023) and Vicuna (Zheng et al., 2023) as the instruction fine-tuned models, using LLaMA (Touvron et al., 2023) as the corresponding pre-trained model[4]. During generation, we employ a greedy search (for reproduction) to generate up to 300 tokens for each input prompt.

**Instruction Datasets.** We collect user-oriented prompting texts from three publicly available datasets: Self-Instruct (Wang et al., 2022), LIMA (Zhou et al., 2023), and MT-Bench (Zheng et al., 2023). The Self-Instruct dataset includes 252 pairs of prompts and responses written by humans, used both for generating more pairs and as a test set. LIMA, mainly based on questions and answers from online platforms like Stack Exchange, has 1000 training pairs and 300 testing pairs. On the other hand, MT-Bench, intended only for machine evaluation, has 80 human-written pairs across eight categories but lacks a training set. Our analysis focuses exclusively on the test sets from these datasets.

## 4 IMPACT OF USER PROMPTS TO INSTRUCTION TUNED MODELS

### 4.1 VISUALIZING PROMPT INFLUENCE ON GENERATION PROCESS

#### 4.1.1 TOOL: QUANTIFY IMPORTANCE BETWEEN INPUT AND OUTPUT TOKENS

We aim to measure the importance of each input token to each response token. In classification, input feature importance is typically measured by monitoring confidence changes upon its removal (Ribeiro et al., 2016; Ebrahimi et al., 2017; Feng et al., 2018). Treating text generation as word classification tasks, the importance of an input to an output token is gauged by examining confidence changes in output generation without the input token. We define *importance* $I_{n,m}$ of input $x_n$ to output $y_m$ as:

$$I_{n,m} = p(y_m|Z_m) - p(y_m|Z_{m,/n}),\qquad(1)$$

where $Z_m$ is the context to generate $y_m$ by concatenating the inquire $X$ and the first $m-1$ tokens of response $Y$, $Z_{m,/n}$ omits token $x_n$ from $Z_m$, and $p(\cdot|\cdot)$ is the conditional probability computed by language model $f$. We accelerate Equation 1 with the first-order approximation: $I_{n,m} \approx \frac{\partial f(y_m|Z_m)}{\partial \mathbf{E}_i[x_n]} \cdot \mathbf{E}_i[x_n]^\top$, where $\mathbf{E}_i[x_n]$ is the input word embedding of token $x_n$ (check Appendix A for theoretical justification). The importance of input tokens cannot be compared across different output tokens due to its dependency on the confidence $f(y_m|Z_m)$. It's crucial to recognize that a word with a lower confidence doesn't necessarily imply it is a trivial word. Specifically, in language modeling, the likelihood of a word $y$ given previous context $x$ could be extended with Bayes' theorem as $p(y|x) \propto p(x|y) \cdot p(y)$. Here, semantic (non-trivial) words have a lower prior probability $p(y)$ since they are less common in the general corpus. In addition, models tend to estimate a lower conditional probability $p(x|y)$ since it is more challenging to predict such meaningful words unless they show very strong semantic relations. Consequently, models are typically more confident about common, less meaningful words, and less confident about semantically rich, rare words. Therefore, we propose to rescale the importance scores derived from the same output token to ensure they are comparable across different output tokens. In addition, we introduce a sparse operation over the rescaled importance to overlook the noise introduced by first-order approximation. To this end, the normalized pairwise score $S_{n,m} = \text{ReLU}\left(\left\lceil L \times \frac{I_{n,m}}{\max_{n'=1}^{N} I_{n',m}}\right\rceil - b\right)$, where $\lceil\cdot\rceil$ is the ceiling function, and $b \in [0, L]$ is a hyper-parameter determining the minimal interested importance level.

#### 4.1.2 EXPERIMENT DESIGNS

This qualitative experiment aims to demonstrate how input prompt words contribute to the response generation via visualizing salient maps based on normalized pairwise importance $s_{n,m}$. We set $L = 10$ and $b = 0$ to faithfully present all information (including noise) for visualization. Figure 1 provides a pair of salient maps to the same prompt corresponding to the model-generated responses from LLaMA and Vicuna, respectively. We show more visualization cases in Appendix C.

---

[4]We implement these models with the code and checkpoints available from Huggingface library (Wolf et al., 2019). We use lmsys/vicuna-7b-delta-v1.1 for Vicuna and tatsu-lab/alpaca-7b-wdiff for Alpaca.

### 4.1.3 EXPERIMENT RESULTS

**Obs-1: Instruction tuning helps the models distinguish between instruction and context words more accurately.** We provide a visualization case that asks the models to classify the tone (instruction) of a given email (context) into one of the listed categories (background). Both models begin their responses by repeating the email. Later, Vicuna successfully follows the human instruction by generating analysis on the tone of the email, while LLaMA fails to follow by saying "I am not sure what you are asking. What is the input?" Figure 1 (right) shows that the instruction part is generally brighter than the background and context part, indicating the strong influence of instruction words in shaping response generation. In contrast, context lines only light up in specific spans and show up a diagonal pattern at the left button of both figures (models are repeating the email), revealing the

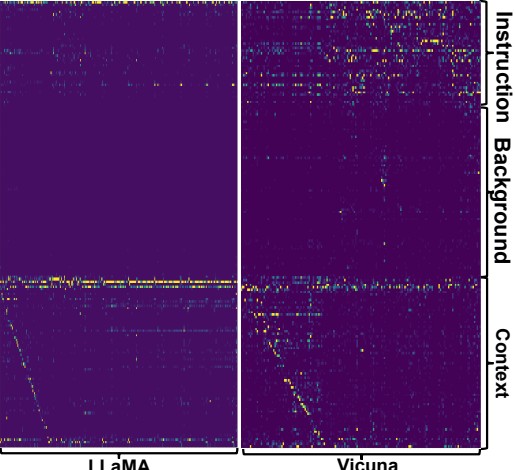

Figure 1: Salient maps of the prompt-response pair[5] from LLaMA (left) and Vicuna (right).

"match-and-reference" behavior. The differences between the left and right plots further highlight the impact of instruction tuning. Specifically, the left plot has certain context lines that appear less bright in the right plot, while certain instruction lines in the right plot stand out more. This visualization case raises a hypothesis that the instruction words *constantly* contribute to the response generation if the model successfully follows the user intention. Sec. 4.2 will quantitatively verify this assumption.

## 4.2 ASSESSING INSTRUCTION FOLLOWING WITH IMPORTANCE DENSITY

### 4.2.1 TOOL: ATTRIBUTE INPUT TOKENS FOR ENTIRE GENERATION PROCESS

We aim to measure the overall attribution of each input token to the entire response generation process. Based on our observations in Sec. 4.1.3, an input token should acquire a greater attribution score if it is important to generate more output tokens. Following this intuition, the input token $x_n$'s attribution $a_n$ is measured by leveraging $\ell_1/\ell_p$ density function over the normalized importance to all output tokens: $a_n = ||S_n||_1/||S_n||_p$, where $S_n = [S_{n,1}, ..., S_{n,M}]$, and $p \in \mathbb{R}^+$ serves as a hyper parameter. One of the nice properties of using $\ell_1/\ell_p$ density function is if two input tokens have the same total sparse importance, then the one having greater maximum importance would receive more attribution score (check (Hurley & Rickard, 2009) for proof).

### 4.2.2 EXPERIMENT DESIGNS

This experiment quantitatively justifies the assumption observed from Sec. 4.1.3 that a model aligns with human intention if it constantly uses instruction words to guide the generation. Specifically, we manually annotate a dataset, where each prompt has been marked its instruction part, and each response is labeled as either "followed" or "unfollowed". Please check Appendix B.1 for the annotation details. Here, the instruction part includes sentences that describe background information and actions for a task. On the other hand, "followed" indicates that the model provides information pertinent to the user intention, regardless of the response's factual correctness. For each prompt-response pair sourced from our datasets, we compute the importance density score with $L = 10$, $b = 7$, and $p = 4$. The hyper-parameters are manually tuned. We further normalize the scores to ensure comparability across different instances and remove the instances with a short response (less than 5 tokens) as their estimations of density are not stable. Table 1 presents the average importance density scores for both followed and unfollowed instances that Vicuna generates the responses. Please check Appendix B.2 for an analysis of outlier cases to these cases.

---

[5]The prompt boldfaces its direct **instruction words** and underlines its background: Analyze the word choice, phrasing, punctuation, and capitalization in the given email. **How may the** writer of this email sound to the reader? These tones include Disheartening, Accusatory, Worried, Curious, Surprised, Disapproving, Unassuming, Formal, Assertive, Confident, Appreciative, Concerned, Sad, Informal, Regretful, Encouraging, Egocentric, Joyful, Optimistic, and Excited.\n\nInput: Hi Jen, \nI hope you're well. Can we catch up today? I'd appreciate your input on my presentation for tomorrow's meeting. I'd especially love it if you could double-check the sales numbers with me. There's a coffee in it for you!\n\nOutput:

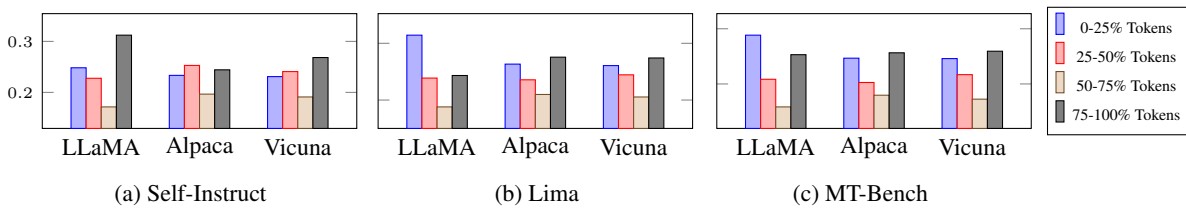

Figure 2: Distribution of importance density over different parts of prompt tokens.

### 4.2.3 Experiment Results

**Obs-2: The importance density on instruction words reflects the models' behaviors in following user intentions.** From Table 1, it becomes evident that attribution scores for "followed" instances consistently outperform those of "unfollowed" across all datasets. This distinction is sta-

Table 1: Importance density on instruction words over followed and unfollowed instances from Vicuna.

| Dataset | Followed ↑ | Unfollowed ↓ | p-value ↓ |
|---|---|---|---|
| Self-Instruct | $1.2964_{\pm 0.51}$ | $0.9124_{\pm 0.46}$ | $1.3e^{-5}$ |
| LIMA | $1.6519_{\pm 0.49}$ | $1.3277_{\pm 0.46}$ | $6.9e^{-5}$ |
| MT-Bench | $1.4639_{\pm 0.56}$ | $0.9295_{\pm 0.51}$ | $3.0e^{-4}$ |

tistically validated by notably low p-values, where the null-hypothesis is the average importance densities of followed and unfollowed instances are equal. Table 1 underscores the strong correlation between the importance density scores of instruction words and the instruction following capability. Case studies in Appendix B.2 suggest that instruction tuned models may pretend to follow instructions without realizing user instructions. Appendix B.1.1 shows that Vicuna achieves greater importance density scores compared to LLaMA across the three datasets, indicating instruction tuning empowers the pre-trained model in better identifying and harnessing instruction words from user prompts.

## 4.3 Exploring Prompt Position with Importance Density

### 4.3.1 Experiment Designs

Each input prompting text from our datasets is divided into individual sentences, with each sentence further split into four same-length segments. We normalize the density scores for a sentence by dividing by their sum and then accumulating them for each segment. The averaged attribution proportions for each segment within the input sentences are depicted in Figure 2.

### 4.3.2 Experiment Results

**Obs-3: Instruction fine-tuned models still overlook the middle and tail of input prompts, but less so than pre-trained models.** Figure 2 shows the importance density score distributed on different segments of input sentences. Both pre-trained and fine-tuned models reveal a notable "U"-shape across all datasets. This is also known as "lost in the middle" (Liu et al., 2023), where they show that SOTA models can overlook central inputs. Unlike their focus on machine reading comprehension, our analysis is grounded on our importance density score on diverse prompting texts, suggesting that this issue commonly and intrinsically exists. When comparing pre-trained to fine-tuned models, we spot a sharper "U" in the former, which becomes less obvious after instruction tuning.

## 5 Evolution of Knowledge in Feed-forward Networks

### 5.1 Tool: Interpret Feed-forward Networks with "Concept" Description

We aim to interpret the knowledge of feed-forward networks in the *concept* level. We treat each feed-forward network $\sigma(\mathbf{X}\mathbf{W}_u^\top)\mathbf{W}_p$ as key-value memories (Geva et al., 2020), where each row vector of $\mathbf{W}_u$ and $\mathbf{W}_p$ stores a textual pattern. However, these textual patterns (neurons) are usually polysemantic (Elhage et al., 2022; Bricken et al., 2023), causing each textual pattern not to be interpreted with a concise meaning. Thus, we propose to seek a set of orthogonal vectors that capture the major directions in which these patterns spread. Formally, given the patterns stored in $\mathbf{W}_p$, we construct the covariance matrix as $\mathbf{C} = \widetilde{\mathbf{W}}_p^\top \widetilde{\mathbf{W}}_p$, where $\widetilde{\mathbf{W}}_p$ is the centralized matrix of $\mathbf{W}_p$ with zero mean of each column. Then the orthogonal basis vectors $\mathbf{V}$ of these patterns satisfy:

$$\mathbf{C}\mathbf{V} = \mathbf{\Lambda}\mathbf{V}, \tag{2}$$

where each column vector of $\mathbf{V} \in \mathbb{R}^{D \times D}$ is unit length, $\mathbf{\Lambda} = diag([\lambda_1, ..., \lambda_D])$, and $\lambda_1 \geq ... \geq \lambda_D \geq 0$. In this context, our primary focus lies on the top-$R$ values of $\mathbf{\Lambda}$ along with their corresponding column vectors in $\mathbf{V}$. This is due to the fact that they show the *principal directions* of the encoded

patterns from $\mathbf{W}_p$. We then project each principal vector to the word embedding space $\mathbf{E}_o$ and find the top-$K$ relevant words for interpretation: $\mathcal{E}_r = \arg max_{\mathcal{V}' \in \mathcal{V}, |\mathcal{V}'| = K} \sum_{w \in \mathcal{V}'} \mathbf{V}^\top [r] \mathbf{E}_o[w]$, where $\mathbf{V}^\top[r]$ is the $r$-th column vector of $\mathbf{V}$, $\mathbf{E}_o[w]$ is the output word embedding of $w$. Since $\mathbf{v}_r$ is a unit vector, $\mathbf{V}^\top[r]\mathbf{E}_o[w]$ measures the projection length of the word vector in this direction. Thus, it is natural to represent this vector with the words having the largest projection length, and the word list could be further summarized as a textual description by a human or a machine annotator.

## 5.2 EXPERIMENT DESIGNS

Before applying our method, we create a new vocabulary derived from ShareGPT (RyokoAI, 2023) to make the candidate words $\mathcal{V}$ more understandable compared to a large number of sub-tokens from the build-in LLaMA vocabulary. We then analyze the first 300 basis vectors of each feed-forward network from LLaMA and Vicuna with their top 15 relevant words. ChatGPT (gpt-3.5-turbo) is considered as our machine annotator for this experiment. Table 2 provides sample word lists and their descriptions, and more cases are available in Appendix E.2. The detailed settings and statistics of concept descriptions are shown in Appendix D.2. Appendix E.1 provides additional discussions to the accumulated explained variance of the decomposed principal vectors.

To study the knowledge evolution, we condense tasks from previous research (Zheng et al., 2023; Ouyang et al., 2022) to scenarios including writing, math, coding, and translation. We then identify which scenarios a concept could be used for (see Appendix D). Note that some concepts may fit multiple scenarios. Also, we sort concepts into phonology [6], morphology [7], syntax, or semantics linguistic levels based on the disciplines in the linguistic subject (Thomas, 2005). Table 3 displays the percentage of knowledge for different scenarios and linguistic levels.

## 5.3 EXPERIMENT RESULTS

**Obs-4: The principal vectors of feed-forward network neurons provide concept-level understandings of the encoded knowledge.** We select 5 representative principal components and their explanations from the last feed-forward network of Vicuna and display them in Table 2. More interpretation cases are available in Tables 10 and 11. Appendix D.2 shows that around 60% of the first 300 principal components from the middle layers of Vicuna could be interpreted by ChatGPT, and reports the statistical results of the word frequency from the machine-annotated descriptions. In Table 2, the descriptions of five principal vectors span a diverse array of topics, ranging from medical ("medical abbreviation") to linguistic ("starting with *the*"). Notably, the concept of medical abbreviations stands out, as it's often difficult for human annotators to discern their medical relevance. This indicates the advantage of utilizing machine annotators for their vast knowledge.

Table 2: Interpret the last feed-forward network of Vicuna with its principal vectors using the word lists and concept descriptions.

| Rank | Description | Words |
|---|---|---|
| 6 | medical abbreviations | CBT, RTK, RT, RH, HRV, MT, HB, PH, PT, ... |
| 8 | starting with "the" | the, theological, theology, theater, thead, ... |
| 11 | programming tasks or actions | provide, nltk.corpus.stopwords.words, sklearn.metrics, ... |
| 26 | Hyphenated terms | a-, one-of-a-kind, state-of-the-art, one-on-one, ... |
| 69 | numbers | sha256, tt, 8266, 768, 1986, 1968, 638, 54a, 86, ... |

**Obs-5: Instruction tuning shifts the principal vectors of feed-forward network neurons toward user-oriented tasks without moving them across linguistic levels.** We observe from Table 3 that Vicuna encodes more concepts than LLaMA for writing, coding, and math tasks, with the difference in writing and coding being statistic significant ($p < 0.1$), where the null-hypothesis is knowledge proportions of a certain category for Vicuna and LLaMA are equal. However, that of concepts for translation is reduced after fine-tuning, indicating multi-linguistic knowledge is forgotten. Although we could observe the changes over the user view, from the linguistic view, it remains the same. In

Table 3: Concept distribution over different user-oriented scenarios and linguistic levels.

| | Category | Vicuna | LLaMA | p-value |
|---|---|---|---|---|
| Scenarios | Writing | $53.05_{\pm 0.46}$ | $51.47_{\pm 0.92}$ | 0.0154 |
| | Coding | $29.45_{\pm 0.43}$ | $28.64_{\pm 0.48}$ | 0.0350 |
| | Math | $5.21_{\pm 0.36}$ | $5.04_{\pm 0.33}$ | 0.5193 |
| | Translation | $25.30_{\pm 0.39}$ | $26.27_{\pm 0.70}$ | 0.0411 |
| Linguistic | Phonology | $1.18_{\pm 0.11}$ | $1.15_{\pm 0.07}$ | 0.6251 |
| | Morphology | $17.16_{\pm 0.49}$ | $16.83_{\pm 0.60}$ | 0.4223 |
| | Syntax | $7.16_{\pm 0.31}$ | $7.52_{\pm 0.50}$ | 0.2551 |
| | Semantic | $74.70_{\pm 0.65}$ | $74.66_{\pm 0.67}$ | 0.9394 |

---

[6]Phonology studies sound systems, e.g. words with *"le" sound*: brittle, tackle, chuckle, pickle.

[7]Morphology studies word structure, e.g. words with *"sub-" prefix*: subarray, subculture, subway.

particular, Vicuna and LLaMA show nearly identical distributions across the four linguistic levels. None of them are statistically significant ($p > 0.1$). This observation suggests instruction tuning does not alter the distribution of concepts across linguistic levels.

**Obs-6: The proportion of semantic knowledge first increases then decreases from bottom to top layers, while that of morphology knowledge does the opposite.** Figure 3 displays how concepts from various linguistic levels are spread across layers. First, there isn't a noticeable distribution shift between Vicuna and LLaMA, which matches Obs-5. One noteworthy observation is the opposite "U"-shape trend for semantic knowledge, mirrored by a regular "U"-shape for morphology. This pattern is somewhat surprising, especially since previous studies in computer vision suggest that (low-level) basic features are extracted in the bottom layers, and (high-level) compositional knowledge is learned in the top layers (Zeiler & Fergus, 2014; Selvaraju et al., 2016). However, since LLaMA is designed

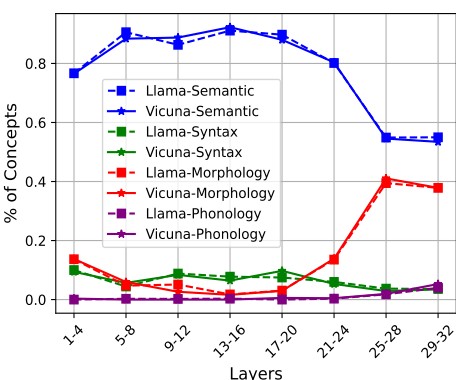

Figure 3: Concept distribution of linguistic levels over different model layers.

for predicting the next words, this unusual trend makes some sense. Specifically, we conjecture that LLaMA learns more morphology knowledge (e.g., prefix and suffix patterns) in the last few layers for simulating a prefix-tree structure (Fredkin, 1960; Giancarlo, 1995; Paladhi & Bandyopadhyay, 2008; Shan et al., 2012). By doing so, LLaMA could use fewer parameters to memorize more word collocations to complete the next-word prediction task. We leave explorations as future work.

# 6 EVOLUTION OF KNOWLEDGE IN SELF-ATTENTION HEADS

## 6.1 TOOL: INTERPRET SELF-ATTENTION HEADS WITH WORD-WORD PATTERNS

We aim to interpret the behaviors of self-attention heads with word pairs. Given a self-attention head, the relation between a pair of words $(w_a, w_b)$ for $w_a, w_b \in \mathcal{V}$ could be approximated by $\mathbf{A}_{a,b} \propto \sum_d^D \mathbf{E}_i[w_a] \mathbf{W}_q^{h\top}[d] \times \mathbf{E}_i[w_b] \mathbf{W}_k^{h\top}[d]$, indicating that the relation $\mathbf{A}_{a,b}$ linearly relates to the activations of column vectors (neurons) of weights $\mathbf{W}_q^h$ and $\mathbf{W}_k^h$. Therefore, we interpret the behavior of a self-attention head by aggregating the word-pairs activated by its neuron-pairs. Specifically, we first interpret neurons $\mathbf{W}_q^{h\top}[d]$ and $\mathbf{W}_k^{h\top}[d]$ by collecting the top-$K$ words that could most activate them, i.e. $\mathcal{E}_q^d = \arg max_{\mathcal{V}' \subseteq \mathcal{V}, |\mathcal{V}'|=K} \sum_{w \in \mathcal{V}'} \mathbf{E}_i[w] \cdot (\mathbf{W}_q^{h\top}[d])^\top$ and $\mathcal{E}_k^d = \arg max_{\mathcal{V}' \subseteq \mathcal{V}, |\mathcal{V}'|=K} \sum_{w \in \mathcal{V}'} \mathbf{E}_i[w] \cdot (\mathbf{W}_k^{h\top}[d])^\top$. We then form word pairs cross $\mathcal{E}_q$ and $\mathcal{E}_k$ by connecting the words having a high probability of appearing within the same context, since the self-attention mechanism is designed to capture word relations within the input texts. Practically, we approximate this probability by computing the cosine similarity of their GloVe (Pennington et al., 2014) word embeddings, which has been proved as a decomposition of word-word co-occurrence matrix $\mathcal{E}_{qk}^d = \{(w_q, w_k) : cos(e_q, e_k) > \theta\}$, where $w_q \in \mathcal{E}_q^d$, $w_k \in \mathcal{E}_k^d$, $\mathbf{e}_q$, $\mathbf{e}_k$ are their GloVe embeddings, and $\theta$ is a threshold. Finally, the behavior of a self-attention head is described with frequent word-pairs activated by its neurons.

## 6.2 EXPERIMENT DESIGNS

We isolates the top-100 most activated words for neurons $\mathbf{W}_q^h[d]$ and $\mathbf{W}_k^h[d]$). From there, we form word pairs with a dynamically determined threshold $\theta$. This threshold is established by calculating the cosine similarity between a word and the 1000 most frequent words with GloVe word embeddings. We then derived the mean and standard deviation of these cosine similarities for each word. The threshold for a given word was set at its mean plus 1.96 times its standard deviation. Once the word pairs of neurons are obtained, the self-attention head's behavior is described with word pairs recognized by at least two of its neuron pairs. We show some examples in Table 12 and Table 13.

We conduct two analyses based on these word pairs. First, we tract alterations in word pairs after instruction tuning, represented by the intersection rate $M = \frac{\mathcal{E}_{pt} \cap \mathcal{E}_{ft}}{\mathcal{E}_{pt} \cup \mathcal{E}_{ft}}$, $\mathcal{E}_{pt}$ and $\mathcal{E}_{ft}$ denote the top-100 word pairs of pre-trained and fine-tuned models. Figure 4 visualizes the change rate $1 - M$ across neuron pairs and heads over various layer groups. Furthermore, we investigated shifts in instruction verbs (e.g., "write", "create", and "classify") within the self-attention heads. We identify

34 instruction verbs based on the statistics of common use cases (Wang et al., 2022; Ouyang et al., 2022). We also assemble a control set of 1000 frequent verbs from general English corpus (Speer, 2022). For each verb, we count the proportion of self-attention heads transitioning from non-inclusion to inclusion of the certain verb after tuning, and report the results in Table 4.

### 6.3 EXPERIMENT RESULTS

**Obs-7: Instruction tuning marginally changes the behaviors of self-attention neurons, but significantly modifies self-attention heads.** Figure 4 shows that as layer depth increases, the rate of change for word pairs related to neurons and heads also increases. Specifically, the change rate for neuron word pairs goes from 20.52% to 22.99%, while for head levels it fluctuates more, going from 30.95% to 35.81%. The change rate for neurons is relatively low, around 20%, but for self-attention heads, it varies more, between 30% and 37%.

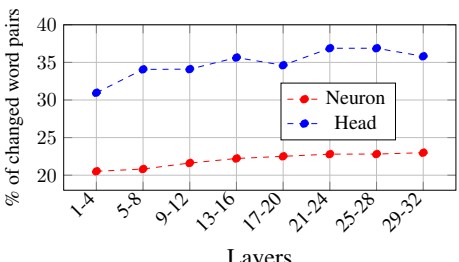

Figure 4: Shift of word-word patterns for self-attention after instruction tuning.

**Obs-8: Instruction tuning encourages lower self-attention heads to encode more word-word patterns related to instruction verbs.** Table 4 shows that more self-attention heads from lower (1-8) and middle (8-24) layers encode word-word relations with instruction verbs after instruction tuning, where the layer groups 1-8 and 9-16 reach statistic significance ($p < 0.1$),

Table 4: Percentage of self-attention heads encoding certain verbs after instruction tuning.

| Layers | Instruct | General | p-value |
|--------|----------|---------|---------|
| 1-8 | $28.78_{\pm25.36}$ | $14.53_{\pm15.87}$ | 0.0180 |
| 9-16 | $23.70_{\pm21.64}$ | $14.03_{\pm13.71}$ | 0.0540 |
| 17-24 | $19.02_{\pm17.47}$ | $15.29_{\pm13.74}$ | 0.3535 |
| 24-32 | $10.91_{\pm9.19}$ | $14.63_{\pm13.92}$ | 0.0781 |

where the increasing percentages of self-attention heads encoding word-word relations with instruct and general verbs are equal. In particular, about $28.78\%$ of the heads from 1-8 layers transits from not encoding instruction verbs to encoding instruction verbs, nearly double the $14.53\%$ of that encodes new general verbs after instruction tuning. Overall, averaging over $20\%$ self-attention heads that encode more instruction relations, only about $14\%$ encode more general verbs. This difference indicates that instruction tuning teaches self-attention to identify various detailed instructions.

## 7 DISCUSSION

Our findings provide a unique perspective to align with recent studies. 1) The importance of *prompt diversity* is highlighted by both us and Zhou et al. (2023); Wang et al. (2022). Since our three findings suggest that instruction tuning links the pre-trained model to user tasks, we could expect a better alignment with human intentions if the model is exposed to broader prompts. 2) The efficacy of *training self-attention with first priority* (LoRA fine-tuning) (Taori et al., 2023; Juletx, 2023) is corroborated by Finding-1 and Finding-3. Specifically, Finding-1 illustrates the capability to distinguish instruction words is essential to the instruction following, while Finding-3 highlights that self-attention heads directly learn instruction knowledge. 3) The advantage of *training feed-forward networks* (fully fine-tuning) (Sun et al., 2023) is evident from Finding-2 and Finding-3, which demonstrate that feed-forward networks update their knowledge toward user tasks.

Our findings also pose three open questions: 1) Can we use the importance density score as a training object in instruction tuning to reflect the instruction following capability inherently? 2) Do LLMs simulate a prefix tree in their upper layer for efficient decoding? 3) How do self-attention modules and feed-forward networks collaborate to generate a helpful response for the user?

## 8 CONCLUSION

This paper presents an inherently comprehensive analysis of instruction tuning for user intention alignment by quantitatively and qualitatively comparing the interpretations between pre-trained and fine-tuned models. Our findings indicate that instruction tuning links the pre-trained model to user intentions, including encoding more instruction words' knowledge within self-attention, and rotating general knowledge from feed-forward networks towards user usage. It is worth mentioning that the interpretability toolbox used in this study can also support future general research on LLMs.

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

## A  PROOF OF LINEARLY APPROXIMATION TO IMPORTANCE SCORES

We prove that equation $I_{n,m} = p(y_m|Z_m) - p(y_m|Z_{m,/n}) \approx \frac{\partial f(y_m|z_m)}{\partial \mathbf{E}_i[x_n]} \cdot \mathbf{E}_i[x_n]^\top$ with the first-order Taylor extension. $p(y_m|Z_m)$ is written as $f(y_m|\mathbf{Z}_m)$, where $f$ is the language model, $\mathbf{Z}_m \in \mathbb{R}^{(N+m-1)\times d}$ are the word embeddings of the input token sequence $Z_m = [x_1, ..., x_N, y_1, ..., y_{m-1}]$, and the $d$-dimensional word embeddings of a token $w \in Z_m$ is defined as $\mathbf{E}_i[w]$. Thus, we first have $I_{n,m} = f(y_m|\mathbf{Z}_m) - f(y_m|\mathbf{Z}_{m,/n})$, where we let the $n$-th row vector of $\mathbf{Z}_{m,/n}$ be zeros.

The first-order Taylor expansion of $f(y_m|\mathbf{Z}_m)$ around $\mathbf{Z}_{m,/n}$ is

$$f(y_m|\mathbf{Z}_m) \approx f(y_m|\mathbf{Z}_{m,/n}) + \left.\frac{\partial f(y_m|\mathbf{Z}_m)}{\partial \mathbf{Z}_m}\right|_{\mathbf{Z}_{m,/n}} \cdot (\mathbf{Z}_m - \mathbf{Z}_{m,/n})^\top.$$

Since the difference between $\mathbf{Z}_{m,/n}$ and $\mathbf{Z}_m$ is the $n$-th row, the term $\mathbf{Z}_m - \mathbf{Z}_{m,/n}$ is just the vector $\mathbf{E}_i[x_n]$. Therefore, the above equation could be simplified as:

$$f(y_m|\mathbf{Z}_m) \approx f(y_m|\mathbf{Z}_{m,/n}) + \frac{\partial f(y_m|\mathbf{Z}_m)}{\partial \mathbf{E}_i[x_n]} \cdot \mathbf{E}_i[x_n]^\top.$$

Bring this approximation to the definition of $I_{n,m}$, we have $I_{n,m} \approx \frac{\partial f(y_m|\mathbf{Z}_m)}{\partial \mathbf{E}_i[x_n]} \cdot \mathbf{E}_i[x_n]^\top$.

## B  ANALYZING IMPORTANCE DENSITY

### B.1  EXPERIMENT SETTINGS

For each collected prompting text from the three public datasets, we let Vicuna and LLaMA generate its corresponding response (Sec. 3.2); we then manually identify the instruction sentences from each input prompt and annotate whether the response provides helpful information ("followed") or not ("unfollowed").

**Annotate instruction and context.** Specifically, the instruction usually describes the user intention with some background (optional), which could be both very long [8] or very concise [9]. Note that we annotate the instruction words on the sentence level, and the template words as "Input:" and "Output:" are not considered. For some prompts, the instruction words may be distributed in both the head and tail of the input text, and we will consider them together. Among these instruction sentences, we define the rest of the input prompt as context words, which is unnecessary to the input prompting text.

**Annotate Followed or Unfollowed Response** We consider the *helpfulness* of the response as the ability of instruction following described by Ouyang et al. (2022). Therefore, if a response is helpful to the user, then we label it with "followed". Specifically, we consider four levels of helpfulness: L1 - the model is randomly saying something or just repeating itself; L2 - the model provides some information that could be used to answer the question, but the model fails to organize it well; L3 - the model generates a response that generally follows the prompts, but missing some detailed instructions; L4 - the response is perfect as a human response. In our study, we consider the responses from L2 to L4 as "followed". Note that we are not concerned about hallucination issues in our study.

#### B.1.1  COMPARING IMPORTANCE DENSITY BETWEEN VICUNA AND LLaMA

Table 5 reports the average importance density over the instruction words by giving different responses generated by Vicuna or LLaMA. We could observe that Vicuna constantly assigns denser importance scores on the instruction words compared to LLaMA across the three datasets, where

Table 5: Importance density on instruction words over responses generated by Vicuna and LLaMA.

| Dataset | Vicuna ↑ | LLaMA ↓ | p-value ↓ |
|---|---|---|---|
| Self-Instruct | $1.2256_{\pm 0.52}$ | $0.9410_{\pm 0.46}$ | $< 1e^{-10}$ |
| LIMA | $1.5946_{\pm 0.51}$ | $1.3048_{\pm 0.45}$ | $< 1e^{-10}$ |
| MT-Bench | $1.3492_{\pm 0.60}$ | $1.2216_{\pm 0.45}$ | $0.1301$ |

---

[8]A long instruction: "How do social media platforms influence the way people consume and share news, and what are the potential implications for the spread of misinformation?"

[9]A short instruction: "to English:"

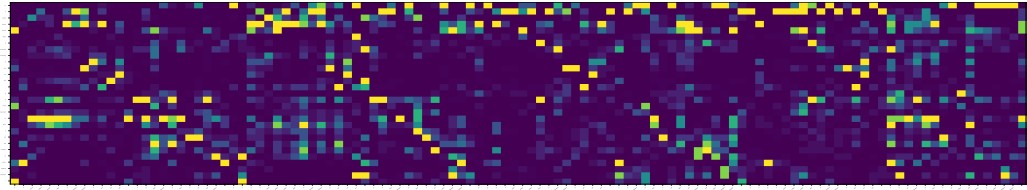

Figure 6: **CaseID:** LIMA-144 **Prompt:** In a world where automation is becoming increasingly prevalent, is it more important to prioritize job creation or technological progress? **Vicuna:** In a world where automation is becoming increasingly prevalent, it is more important to prioritize job creation.\n\nIn a world where automation is becoming increasingly prevalent, it is more important to prioritize technological progress.\n\nIn a world where automation is becoming increasingly prevalent, it is important to prioritize both job creation and technological progress.\n\nIn a world where automation is becoming increasingly prevalent, it is not necessary to prioritize either job creation or technological progress.

this improvement is validated by student-t-test, where the null hypothesis is that the average importance densities computed by responses generated by Vicuna and LLaMA are equal. According to the conclusion claimed in Sec. 4.2.3 that a higher importance density on instruction words indicates a better instruction following capability, we draw our conclusion that Vicuna could better follow user instructions on the Self-Instruct and LIMA datasets compared to LLaMA, while achieving a marginal improvement on MT-Bench dataset. It is not surprising since the MT-Bench dataset includes instructions that are much more challenging than those of the others.

## B.2 CASE STUDY ON OUTLIERS

Instruction fine-tuned models may pretend to follow the instructions. Figure 5 visualizes a salient map of an instance related to writing enhancement (please see the caption for details). Vicuna's response addresses grammatical errors and modifies sentence structures for improved clarity. A key observation from the figure is that only the first three instruction tokens guide the response generation (Red Box). Specifically, the first three words are "The sentence you", which seems to be not the key instruction verbs like "Rewrite" from the second sentence. Also, some words from the context part are acted as instruction words (Blue Box), which are "\\nInput:" and "\\nOutput:" from the prompt template. These are the words that should be considered as the instruction words since they do not provide the user's intentions. Additionally, a distinctive diagonal line spans the context section, hinting at the model's predisposition to echo context from preceding content. This figure suggests that the model is leveraging inherent language modeling ability rather than identifying and acting upon the given instructions. Thus, we point out that assessing instruction-following abilities based solely on the correlation between input prompts and output responses might not provide an accurate reflection of the model's internal behaviors, while it is still a common strategy to develop the Reward model for RLHF process (Ouyang et al., 2022).

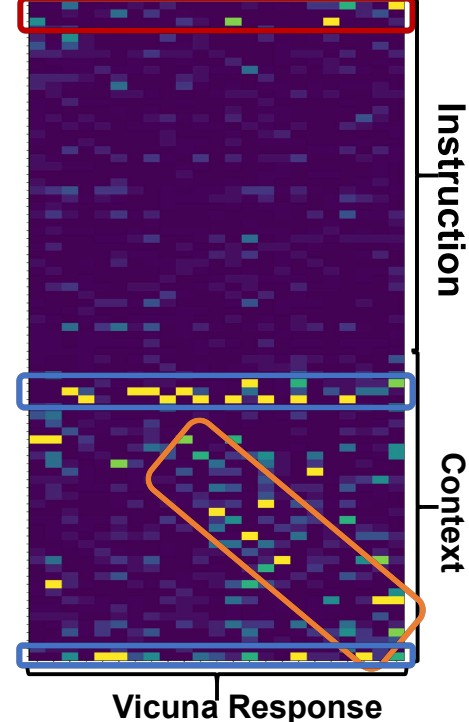

Figure 5: **CaseID:** Self-Instruct-1 **Prompt:** The sentence you are given might be too wordy, complicated, or unclear. Rewrite the sentence and make your writing clearer by keeping it concise. Whenever possible, break complex sentences into multiple sentences and eliminate unnecessary words.\n\nInput: If you have any questions about my rate or if you find it necessary to increase or decrease the scope for this project, please let me know.\n\nOutput: **Vicuna:** Do you have any questions about my rate or do you need to adjust the project scope? Please let me know.

However, we have identified certain instances where our importance density fails. This is predominantly due to our density function's lack of positional awareness. For instance, in Figure 6, the entire user input comprises instruction words. The map suggests that these words play a crucial role in guiding the generation, even towards the latter part of the responses. Under our hypothesis, it would appear the model is following user instructions. Yet, Vicuna seems to merely reiterate the input prompt repetitively, resulting in recurring diagonal patterns. We recommend future research to address this shortcoming, either by adopting a density function that's positionally aware or by integrating a step to identify and handle repetitive responses early on.

## C    VISUALIZING SALIENT MAPS

### C.1    EXPERIMENT SETTINGS

Contrary to the examples shown in the primary content, which utilize golden responses, our focus here is on the connections between user inputs and model outputs. To achieve this, we generate responses from LLaMA and Vicuna, following the protocol laid out in Sec.3.2. Subsequently, we derive the salient maps as per the technique introduced in Sec.4.1.1.

To ensure the maps provide an accurate depiction of the generation process, we set $L = 10$ and $b = 0$. Each map's vertical axis denotes the prompting texts, whereas the horizontal axis symbolizes the generated responses. The intensity of each data point corresponds to the association strength between the respective input and output tokens, with brighter points indicating stronger relationships (visualizing with the best colors).

### C.2    EXPERIMENT RESULTS

Figure 9-14 validate our qualitative assessment that instruction words in user inputs are critical in guiding the generation process. It's evident that each context word typically has limited influence on the response. Collectively, these salient maps underscore the validity of input attribution, achieved by gauging the density of the sparse and normalized importance scores.

## D    SCALING UP WITH AUTOMATED TOOLS

We build upon recent advancements in automated interpretation, using cutting-edge large language models (Taori et al., 2023; Peng et al., 2023; Steven et al., 2022) to emulate human annotators in generating high-level interpretations. By leveraging machine annotators, we could easily scale up our methods to analysis the entire model, providing a more solid results to our findings.

### D.1    EXPERIMENT SETTINGS

**Generating Configuration.** We employ ChatGPT [10] as our machine annotator. Our experiments utilize the gpt-3.5-turbo-0613 model with a hyper-parameter top-$p$=0.9 for nuclear sampling. To mitigate the variability in language model outputs, we repeat the experiment five times. In each iteration, we first condense the top-$K$ words of a specific basis vector into a distinct concept, then pinpoint the user-oriented tasks and linguistic levels associated with these concepts. For our initial interaction with ChatGPT, the temperature is set to 0—signifying a greedy search strategy. In subsequent interactions, we set the temperature to 1. Nevertheless, when identifying tasks and levels, we consistently maintain the temperature at 0.0.

**Prompt Design.** Effective automated interpretation hinges on well-crafted prompts. We meticulously design these prompts using three strategies: role-play, in-context conversational examples, and exclusively high-quality examples.

---

[10]https://platform.openai.com/docs/guides/gpt

*Template-1: Describing words with concise concepts.* The top-15 most activated words coming from the method presented in Sec. 5.1 will be directly appended to this template.

```
System: You are a neuron interpreter for neural networks. Each neuron
looks for one particular concept/topic/theme/behavior/pattern. Look at
some words the neuron activates for and summarize in a single
concept/topic/theme/behavior/pattern what the neuron is looking for.
Don't list examples of words and keep your summary as concise as possible.
If you cannot summarize more than half of the given words within one clear
concept/topic/theme/behavior/pattern, you should say 'Cannot Tell'.

User: Words: January, terday, cember, April, July, September, December,
Thursday, quished, November, Tuesday.
Agent: dates.

User: Words: B., M., e., R., C., OK., A., H., D., S., J., al., p., T., N.,
W., G., a.C., or, St., K., a.m., L..
Agent: abbrevations and acronyms.

User: Words: actual, literal, real, Real, optical, Physical, REAL,
virtual, visual.
Agent: perception of reality.

User: Words: Go, Python, C++, Java, c#, python3, cuda, java,
javascript, basic.
Agent: programing languages.

User: Words: 1950, 1980, 1985, 1958, 1850, 1980, 1960, 1940, 1984, 1948.
Agent: years.

User: Words:
```

*Template-2: Identifying applicable user-oriented tasks.* Summarized concepts are concatenated to this template. We check the writing task into three tasks because ChatGPT often deems nearly every concept suitable for writing. We regard any of these detailed tasks as the primary purpose of writing.

```
System: Which of the following assistant tasks can the given concept
is used for?\n\nTasks: daily writing, literary writing, professional
writing, solving math problems, coding, translation. Return 'None' if it
cannot be used for any of the above tasks. If it could be used for
multiple tasks, list all of them and seperate with ';'.

User: Concept: Words are social media post tags.
Agent: daily writing

User: Concept: Words are Latex code for drawing a grouped barchart.
Agent: professional writing

User: Concept: Words are foreign words or names.
Agent: translation

User: Concept: Words are URLs.
Agent: None

User: Concept: Words are Words related to configuration files and
web addresses.
Agent: coding

User: Concept: Words are rhyming words.
Agent: literary writing

User: Concept: Words are programming commands and terms.
Agent: coding
```

```
User: Concept: Words are
```

*Template-3: Identifying linguistic level.* Any automated summarized concept will be directly concatenated to this template.

```
System: You are a linguist. Classify the provided concept into
one of the following categories: Phonology, Morphology, Syntax,
and Semantic.

User: Concept: Words are dates.
Agent: semantic

User: Concept: Words are perception of reality.
Agent: Semantic

User: Concept: Words are abbrevations and acronyms.
Agent: Morphology

User: Concept: Words are related to actions or activities.
Agent: Syntax

User: Concept: Words are medical abbrivations.
Agent: Semantic

User: Concept: Words are URLs.
Agent: Morphology

User: Concept: Words are verbs.
Agent: Syntax

User: Concept: Words are adjective.
Agent: Syntax

User: Concept: Words are rhyming words.
Agent: Phonology

User: Concept: Words are programming languages.
Agent: Semantic

User: Concept: Words are
```

## D.2 EXPERIMENT RESULTS

Figure 7 illustrates the proportion of word lists that can be induced to a concise concept by our machine annotator. According to our template, if "Cannot Tell" exists in the word list descriptions, we consider that this concept has failed to be interpreted. We have observed that the Vicuna and LLaMA models display comparable levels of interpretability, with no significant distinctions between them. A noticeable trend emerges as the number of layers increases: the ability to explain their encoded concepts improves. Specifically, within layers 24-28, the average

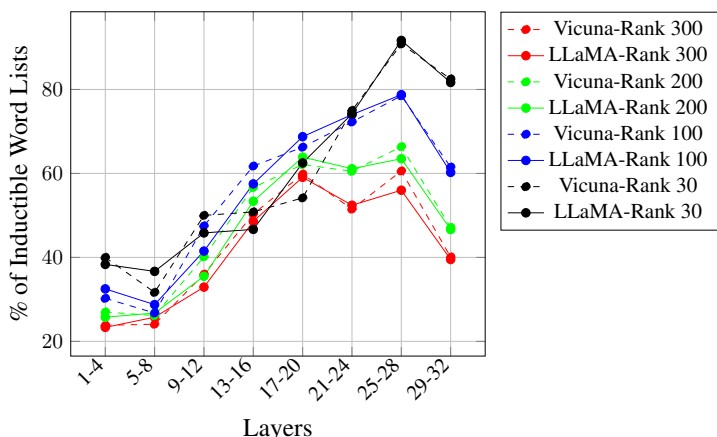

Figure 7: % of represented word lists from top-ranked basis vectors that could be described with a clear concept.

Table 6: Frequency [rank] shift of words from concept description after instruction tuning.

| Layers 1-4 | | Layers 5-8 | |
|---|---|---|---|
| **Frequency↑** | **Frequency↓** | **Frequency↑** | **Frequency↓** |
| language[3] | quality[-3] | programming[0] | foreign-language[0] |
| behavior[83] | describing[-2] | describing[38] | technology[-19] |
| English[79] | characteristic[-1] | computer[11] | Spanish[-33] |
| process[4] | communication[-22] | operation[11] | technical[-32] |
| software-development[8] | something[-18] | computer-science[66] | multilingual[-8] |
| multilingual[64] | start[-43] | development[53] | something[-8] |
| analysis[67] | adjective[1] | language[0] | process[0] |
| operation[33] | foreign-language[-1] | syntax[17] | characteristic[-1] |
| attribute[5] | various[-12] | manipulation[14] | variation[-9] |
| Spanish[14] | concepts/functions[-19] | terminology[22] | functions/methods[-7] |

Table 7: Frequency [rank] shift of words from concept description after instruction tuning. (continued)

| Layers 9-12 | | Layers 13-16 | |
|---|---|---|---|
| **Frequency↑** | **Frequency↓** | **Frequency↑** | **Frequency↓** |
| method[89] | translation[0] | programming[0] | process[-1] |
| french[13] | operation[-31] | software-development[8] | expression[-45] |
| understand[34] | software-development[-17] | language-proficiency[10] | syntax[-5] |
| communication[10] | process[0] | concepts/keys[29] | variation[-15] |
| concepts/functions[41] | foreign-language[0] | terminology[119] | language-related[-24] |
| language-agnostic[23] | programming[0] | language-independent[52] | ambiguity[-49] |
| German[31] | concepts/methods/functions[-61] | concepts/functions[16] | handling[-32] |
| comparison[50] | multilingual[-5] | French[51] | language[0] |
| variety[35] | property[-75] | communication[4] | cultural[-93] |
| technology[28] | language[0] | localization[96] | attribute[-14] |

interpretability rate for the first 30 concepts peaks at 91.67%. This high interpretability rate underscores the effectiveness of our proposed method. It can aptly convey in clear, concise text the knowledge encoded by these models. However, there's a caveat: knowledge encoded closer to the output layer, specifically between layers 28-32, becomes more challenging to elucidate. Interestingly, this particular challenge wasn't present when applying automated interpretation tools to GPT-2 (Millidge & Black, 2022), indicating the behaviors between small and large models are different. Additionally, our findings indicate a decreasing trend in interpretability for concepts that are ranked further back. Overall, these results validate the efficacy of our proposed method in analyzing the knowledge encoded within models.

Table 6-9 enumerates the words that experienced the most significant changes in frequency after instruction tuning, we also show the change of rank following. These words are meaningful words (at least four characters and not a stopword) extracted from the concept descriptions generated by our machine annotator. From the tables, certain words, notably "language", "programming", and "process", displayed significant shifts in frequency after instruction tuning. Linguistic terms ("Spanish", "translation") and technical terms ("method", "programming" and "software") exhibited noticeable changes in various layers. Interestingly, "language" consistently surfaced in almost every layer group, with its frequency both rising and dropping. This observation indicates that different layers are responsible for encoding different categories of knowledge. Specifically, the bottom layers are responsible for storing more general basic knowledge ("behavior", "operation", "adjective"), the middle layers are responsible for learning more abstract knowledge for serving users ("functions/methods", "programming", "software development"), and the higher layers are responsible for learning more knowledge for efficient text generation ("start with", "rhyming", "sound", "letter",). Broadly, the increased mention of words pertinent to user scenarios after finetuning underscores the model's refined focus on user-centric tasks and applications.

Table 8: Frequency [rank] shift of words from concept description after instruction tuning. (continued)

| Layers 17-20 | | Layers 21-24 | |
|---|---|---|---|
| Frequency↑ | Frequency↓ | Frequency↑ | Frequency↓ |
| programming[0] | foreign-language[-2] | manipulation[50] | programming[-2] |
| language[1] | translation[-1] | adjective[9] | state[-12] |
| syntax[78] | variation[-20] | specific[81] | translation[-5] |
| process[2] | expression[-15] | object[42] | quality[-7] |
| language-related[-24] | interaction[24] | adjective[-27] | value[48] |
| time-related[14] | feature[-30] | location[48] | difficulty[-77] |
| language-proficiency[5] | characteristic[-5] | variation[9] | action[0] |
| terminology[123] | duration[-33] | language[1] | prefix[-1] |
| technology[121] | choice[-135] | relationship[121] | start[1] |
| programming-language[-70] | quality | personal[72] | activity[-2] |

Table 9: Frequency [rank] shift of words from concept description after instruction tuning. (continued)

| Layers 25-28 | | Layers 29-32 | |
|---|---|---|---|
| Frequency↑ | Frequency↓ | Frequency↑ | Frequency↓ |
| language[4] | start with[0] | start with [0] | foreign-language[-10] |
| interaction[117] | sound[-1] | sound[4] | language[-3] |
| combination[2] | programming[-1] | rhyming[15] | suffix[-4] |
| variation[1] | action[-2] | combination[9] | abbreviation[-2] |
| software | number[0] | letter[0] | numerical[-5] |
| event[66] | alphanumeric[-23] | process[8] | abbreviations/acronyms[-8] |
| manipulating[53] | abbreviations/acronyms[0] | French[7] | Spanish[-7] |
| operation[28] | pattern[-3] | number[1] | programming[-2] |
| measurement[60] | suffix[-45] | similarity[53] | Indonesian[-18] |
| spell[55] | string[-56] | measurement[43] | sequence[-34] |

# E  INTERPRETING FEED-FORWARD NETWORKS

## E.1  DETAILS OF THE PCA RESULTS

Figure 8 displays the averaging accumulated explained variance of decomposed principal components across the 32 layers, where the translucent area indicates their standard deviations. Since LLaMA and vicuna show almost exactly the same line, we omit LLaMA from this figure. From the figure, we have several observations. Firstly, we find that the accumulated explained variance increases smoothly, where almost half of the basis vectors could explain around 80% of the variances. This observation demonstrates that these neurons do not focus on expressing a few certain features, emphasizing the diversity of the learned hidden features. In addition, the black arrow points out that the accumulated explained variance of the 300 basis vector is about 22.49%, where 300 is the number of basis vectors we studied in this research. It validates that the top 300 parameters are expected to be interpretable since their accumulated explained variance is only 22.49%.

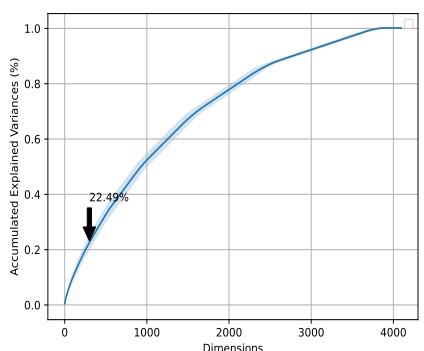

Figure 8: Accumulated explained variance of feed-forward networks from Vicuna.

## E.2  QUALITATIVE ANALYSIS TO INTERPRETABILITY OF PRINCIPAL COMPONENTS

Table 10 and Table 11 list cases that are well interpreted by ChatGPT-turbo-3.5-0613. From these cases, we found that the concept descriptions generally reflect what is behind the word lists well.

## F  INTERPRETING SELF-ATTENTION HEADS

Table 12 and Table 13 list more word pairs for the self-attention heads from the first and the last layers. Typically, these cases are evidence that the extracted word pairs show some insightful relations when we read each one individually. However, when we read them together, it cannot reflect such a concise concept as the feed-forward networks.

Instruction tuning may distill the behaviors of some neurons. We provide one neuron pair that encode a concise concept to study how instruction tuning evolves the neuron behaviors. Here, neuron-pair $(Layer = 31, Head = 24, Dim = 24)$ capture relations in computers (such as backend=authentication, icon=keyboard, GPU=PS, git=curl, and so on). After instruction tuning, the model finds more computer-related word pairs (GPU=motherboard, VM=motherboard, tab=keyboard, mongo=staat, mongo=orden) and overlooks some un-related word pairs (dense=bright, convinced=confused), though the new relations may be not valid. This case is a straightforward evidence that the instruction tuning only re-organizes the knowledge of existing models.

Table 10: Representing words, application scenarios, and linguistic level of the concepts encoded by the 32ed (last) feed-forward network in Vicuna.

| Rank | Scenario | Linguistic | Concept | Top-15 Words |
|---|---|---|---|---|
| 1 | writing; translation | morphology | phrases and abbreviations | everybody, regardless, in, whilst, a.C., vs., amid, I., U.S., Ph.D., anyway, a.m., 6., 9., … |
| 6 | writing; translation | morphology | medical abbreviations | CBT, RTK, RT, RH, HRV, MT, HB, PH, PT, GnRH, HRM, PWV, RS, TB, RL |
| 7 | writing; coding | semantic | URLs and web-related terms | //example.com/data, //example.com/data.json, //www.youtube.com/watch, //image.pollinations.ai/prompt/A, //image.pollinations.ai/prompt/, the, event.target.value, //api.example.com/data, //www.npmjs.com/package/matrixmath, //example.com/api/data, community-based, security-related, industry-specific, //leetcode.com/discuss, //engageleads-gallery.s3.us-west-2.amazonaws.com/ProfitQuotesV2 |
| 8 | writing | semantic | starting with "the" | the, theological, theology, thead, primarily, involving, mainly, theater, alongside, throughout, theatrical, specifically, theta, theorist, regardless |
| 9 | coding | semantic | software development tools and concepts | reCAPTCHA, REST FRAMEWORK, CAPTCHA, ARISTOCRAT, sophistication, REGEXP, sophisticated, PETSC_COMM_WORLD, JEDI_MIND_TRICKS_01, INSTALLED_APPS, ARGFRA, credentials.yaml, OWASP, GARETH, sHSADLH |
| 11 | coding | semantic | programming tasks or actions | provide, nltk.corpus.stopwords.words, sklearn.metrics, install.sh, file.write, serve, give, res.send, clf.fit, pickleball, promote, uint256, giveaway, create, St. |
| 13 | coding | semantic | programming functions/methods | sklearn.feature_extraction.text, re.sub, subprocess.run, a.C., z.string, a.m., e.target.value, request.data.get, p.m., data.length, re.search, f.write, //example.com/data.json, nltk.corpus.stopwords.words, event.target.value |
| 14 | writing; coding | morphology | acronyms and specific terms | sHSADLH, reCAPTCHA, CARRERAS, cv2.COLOR_BGR2GRAY, VXLAN, ARISTOCRAT, OWASP, CAPTCHA, LGBTQ, SOUTHGATE, SARANTIDIS, RTK, RESO, SILVIA, OMENS |
| 16 | math; coding | semantic | number ranges | G-4-8, 5-, 4-, 1-, a-, a-zA-Z0-9, 3-, 2-, 1-5, 5-6, 5-7, 2-5, 4-5, 4-6, 3-5 |
| 21 | math | semantic | numbers | 3,4,5, 2,500, 8,10, 3,500, 75,000, 1,500, 25,000, 6,000, 4,000, 5,000, 7,000, 0,0,0, 8,000, 0,1, 401k |
| 24 | not list above | semantic | characteristics/attributes | health-conscious, learning-based, Content-Security-Policy, Windows-Support-Core, health-related, a-, professional-looking, pay-per-click, Write-Host, user-, cruelty-free, X-Real-IP, energy-efficient, Q-learning, easy-to-use |
| 32 | translation | semantic | foreign languages (possibly Japanese and French) | itu, desu, Deleuze, Shu, baru, meu, -r, atraer, Putu, -u, puddle, sûr, keluar, Veuillez, Meru |
| 35 | writing | phonology | Words with the "le" sound | oooOOOOOooo, ile, brittle, tl, tackle, itle, Isle, Jingle, post-apocalyptic, hl, Michele, tol, preciso, Marlene, needle |
| 42 | translation | semantic | Indonesian words | itu, desu, meu, Thu, baru, vacuum, -u, Shu, satu, Putu, fluctuation, individu, chihuahua, perpetuating, Abu |
| 44 | not list above | semantic | decades | 1960s, 1950s, 1940s, 1970s, 1980s, 1930s, 1920s, 15-minute, 2016/679, 60-minute, 1440p, 755-10145957, 1965, 1963, 1946 |
| 71 | translation | semantic | verbs in different languages | incluyen, weren, soften, brighten, shorten, permiten, behoeften, konten, citizen, teen, willen, Karen, digitalen, starten, crimen |
| 73 | writing | semantic | Words related to "words ending with 'b'" | 4b, bubbly, 1b, rb, -b, Mbappe, childbirth, carb, bulb, herb, heb, Colab, limb, b0, /b |
| 74 | coding | semantic | data types and numbers | 24h, uint256, int256, u32, int32, Kaggle, bytes32, 232, 225, 272822, wag, uh, 32, 23, 325 |
| 75 | writing | morphology | words containing the syllable "jab" | shad, hijab, mariadb, Chad, slab, pedicab, tbsp, jab, scarab, rebound, TaskRabbit, bead, Colab, screech, Abdul-Jabbar |
| 102 | writing; translation | semantic | occupations/jobs | compressor, vraag, destructor, juror, Surveyor, kantor, tremor, effector, flavor, investor, scissor, explorar, projector, escritor, lanjut |

Table 11: Representing words, application scenarios, and linguistic level of the concepts encoded by the 1st (first) feed-forward network in Vicuna.

| Rank | Scenario | Linguistic | Concept | Top-15 Words |
|---|---|---|---|---|
| 1 | writing; translation | morphology | Abbreviations and acronyms | in, I., and, a.C., OK., a.m., U.S., Ph.D., M., B., for, D.C., vs., Feb, to |
| 7 | not list above | semantic | words related to "variability" | architectural, comprise, receivable, usable, Drawable, variability, usability, vanity, circularity, salivary, end=, eget, vise, end, Paradise |
| 25 | writing | semantic | Words related to rhyming | immersing, leer, saber, yer, dreamer, poker, deer, roller, valuing, rester, tracing, Tuner, shower, loser, blocker |
| 27 | coding | semantic | programming concepts | alta, relativedelta, consumed, System.Text, actionable, 'price, payable, island, 'href, belakang, renewable, System.out.println, 'new, 'General, action-oriented |
| 36 | writing | semantic | words related to actions or activities | tingling, Booking, Jingle, citing, bidding, advising, Thing, amazing, CMS, striving, infringement, occurring, Jingdiao, grabbing, fast-growing |
| 37 | writing | morphology | Words related to suffix "-ic" | cystic, Mythic, politic, panoramic, flavorful, opic, antic, physicist, ionic, chronic, employability, effector, spic, silicone, obstructive |
| 38 | writing | semantic | verbs and adjectives related to actions and behaviors | valuing, behaving, sacrificing, advising, environnemental, composing, occurring, encouraging, upbringing, opposing, Bring, petal, charging, arriving, regal |
| 44 | not list above | semantic | qualities or characteristics | bridging, youngest, ±, smartest, brightest, darkest, fullest, pest, comma-separated, celestial, vow, richest, chest, ilmaisee, endow |
| 58 | writing | semantic | verbs related to actions or behaviors | Poe, wee, advocate, relating, moderate, advocating, advocated, tomate, participate, flee, moderating, complexe, anticipate, participating, reiterate |
| 64 | writing | morphology | adjectives with "-ive" suffix | insignificant, restrictive, persuasive, sportive, distinctive, deceive, expressive, decisive, captive, secretive, addictive, defective, digestive, intrusive, abusive |
| 70 | writing | morphology | Words ending in "y" or containing "y" | this.y, flashy, -y, soy, shy, 3y, 2y, toy, Ms., prey, nn.Conv2d, unistd.h, 's3, 's, pre-sale |
| 72 | writing | semantic | adjectives related to characteristics or properties | constitutional, institutional, withdrawal, convolutional, causal, beachten, geral, ethereal, unconstitutional, instrumental, positional, kwaliteit, environmental, incremental, tidal. adjectives related to characteristics or properties |
| 133 | writing | syntax | verbs | lesser-known, lesser, réaliser, dryer, booster, researcher, préparer, uploader, foster, photographer, créer, conocer, fetcher, streamer, minder |
| 135 | writing | syntax | adverbs | substantially, environmentally, latte, surprisingly, weekly, curly, recursively, beautifully, concurrently, texte, confidently, aforementioned, Sally, sadly, honestly |
| 137 | writing | syntax | adverbs | lightly, repurposing, eagerly, frankly, calmly, Polly, preciso, quarterly, analyzing, openly, Thirdly, electrolyte, importantly, shampoo, Secondly |
| 145 | writing | syntax | adverbs and adjectives | environmentally, scientifically, verbally, product_name, conditionally, latest, //www.example.com, Cathy, minimally, socially, Gakkai, /i, modi, annually, accidentally |
| 158 | writing | syntax | adverbs | dispensary, seront, Edmonton, honestly, calmly, unintentionally, supposedly, openly, gracefully, professionally, conditionally, elderly, youngest, infestation, thinly |
| 161 | writing | syntax | adverbs and adjectives | Optionally, conditionally, environmentally-friendly, morally, waving, environmentally, traditionally, Bally, incrementally, emotionally, intentionally, computationally, waking, Ideally, slash |
| 163 | writing | syntax | adverbs and adjectives | heavenly, Plotly, promptly, conveniently, leisurely, vastly, surprisingly, reportedly, brightly, substantially, warmly, équipements, indirectly, falter, elderly |
| 166 | not list above | phonology | language patterns or phonetic similarities | thinner, Skinner, chilies, probably, hypothetical, spinner, SMTP, bietet, så, Jay, witty, seriousness, aforementioned, rapidly, aktif |

Table 12: Most frequent word-pairs activated by self-attention head's neurons from the 32nd (last) layers in Vicuna and LLaMA.

| Head | Keep (%) | Vicuna | LLaMA |
|------|----------|--------|-------|
| 1 | 66.67 | Mann=auf, cloth=trim, prominent=scholar, Morris=Vic, liter=sel, Lake=Montreal, Duke=University, preserved=retrieved, beach=situated, angry=ugly, Chile=Cruz, magnetic=mechanical, assign=restrict, Australian=Russell, administrative=personnel | cloth=trim, prominent=scholar, Morris=Vic, Lake=Montreal, Duke=University, preserved=retrieved, beach=situated, angry=ugly, Chile=Cruz, Cub=RC, magnetic=mechanical, assign=restrict, administrative=personnel, år=œuvre, elem=sau |
| 2 | 76.47 | corn=dry, bullet=triple, delle=seg, mit=nie, Ellen=wife, Collins=Lucy, John=Justin, intent=justify, Amsterdam=Napoli, Ende=bei, broke=broken, Jo=Wang, Amsterdam=Michel, Ri=sou, chr=instanceof | bullet=triple, delle=seg, mit=nie, Ellen=wife, Collins=Lucy, John=Justin, intent=justify, Amsterdam=Napoli, Ende=bei, sobre=votre, broke=broken, Jo=Wang, Ri=sou, chr=instanceof |
| 3 | 57.89 | abort=args, dwell=servant, exc=nur, blue=pale, Jackson=Julia, Henry=Julia, blue=pendant, schon=tra, roof=smoke, p=sl, ahead=wait, meta=todo, ga=ont, Bruce=Graham, Marcus=Martin | abort=args, curve=gradient, predict=prediction, blue=pale, Jackson=Julia, Henry=Julia, blue=pendant, schon=tra, roof=smoke, p=sl, ahead=wait, formally=previously, meta=todo, ga=ont |
| 4 | 57.89 | args=initialization, Elisabeth=Finland, IE=popup, Ferdinand=Sr, IB=Sr, qu=voir, Anderson=Christopher, Pierre=Wright, Christian=Wright, Anderson=Christian, bald=tongue, algebraic=dimensional, approximation=dimensional, Florence=Spring, declared=ruled | args=initialization, concern=significant, confusing=dangerous, Elisabeth=Finland, Ferdinand=Sr, IB=Sr, Anderson=Christopher, Pierre=Wright, Christian=Wright, Anderson=Christian, bald=tongue, beide=λ, algebraic=dimensional, Florence=Milan, approximation=dimensional |
| 5 | 42.86 | clés=terme, dat=sich, Johnson=Wayne, delle=lac, Marx=Wagner, len=unsigned, dei=lac, Anthony=Laura, completed=recently, AC=replacement, peu=ro, sphere=uniformly, examine=explore, ball=throw, referenced=referred | dat=sich, Johnson=Wayne, fi=pu, delle=lac, Marx=Wagner, len=unsigned, dei=lac, completed=recently, le=peu, peu=ro, tomb=tunnel, referenced=referred, Arnold=Carol, Chris=Dave, clue=mistake |
| 6 | 57.89 | ce=va, hal=ord, ord=porta, Jonathan=Walker, divine=mighty, Franklin=Township, Franklin=Nancy, arr=aux, nom=tot, bl=fe, Reich=nur, inequality=labour, entirely=perfectly, az=volt, byte=len | transfer=transmission, ce=va, hal=ord, ord=porta, hun=ja, Jonathan=Walker, Franklin=Township, Franklin=Nancy, ou=voor, nom=tot, Reich=nur, ale=nicht, entirely=perfectly, az=volt, byte=len |
| 7 | 36.36 | screen=touch, divine=virtue, van=ze, creation=divine, Ferdinand=VII, Hamilton=Northern, depth=wide, Marie=Vincent, layer=wire, tak=tut, Mary=Virginia, hitting=scored, laugh=mouth, CNN=Miami, dispute=ownership | divine=virtue, Ali=Campbell, Ferdinand=VII, depth=wide, Marie=Vincent, layer=wire, Mary=Virginia, laugh=mouth, CNN=Miami, agree=doubt, Greece=India, governor=provincial, especial=lugar, Brook=Gabriel, cum=fat |
| 8 | 57.89 | Alice=Robert, Oliver=poet, Sydney=exhibition, quanto=wird, friendly=helpful, ob=typeof, Vater=vida, acid=iron, Jersey=Washington, Anthony=Philip, Kenneth=Robert, Dick=Morgan, hasta=trabajo, gradually=occasionally, press=merge | Oliver=poet, manuscript=photograph, Hong=Sydney, Sydney=exhibition, quanto=wird, friendly=helpful, ob=typeof, Vater=vida, Anthony=Philip, Philip=VI, Kenneth=Robert, Dick=Morgan, hasta=trabajo, gradually=occasionally, Andy=Eric |
| 9 | 66.67 | bin=fi, schon=tre, rib=rub, acc=nur, ch=fi, jam=jar, ob=tre, original=version, brilliant=pleasant, alto=lista, Campbell=Franklin, gegen=hur, Leonard=Oxford, Amy=Glen, Atlanta=Institute | bin=fi, rib=rub, acc=nur, ch=fi, gar=rem, jam=jar, St=Walker, ob=tre, original=version, brilliant=pleasant, alto=lista, Campbell=Franklin, April=Lisa, Leonard=Oxford, Amy=Glen |
| 10 | 50.00 | beach=yard, derive=numerical, Keith=Patrick, dice=ê, Billy=Clark, lin=tym, voltage=wire, Lawrence=Tim, Joseph=Paul, creature=darkness, Palace=XVI, Kelly=Morris, diameter=rectangular, Austin=Johnson, Monument=eigen | beach=yard, derive=numerical, Brook=Keith, Keith=Patrick, dice=ê, Billy=Clark, overlay=width, lin=tym, voltage=wire, autor=bajo, Joseph=Paul, quite=wrong, creature=darkness, hate=miss, Palace=XVI |
| 11 | 66.67 | enjoy=relax, mut=sobre, mouth=tract, dan=mo, pub=situated, excitement=pride, drink=taste, bas=tym, zijn=¡¡, prev=, Irish=Thomas, Bruce=Richard, Helen=Richard, bless=sweet, Ivan=Joan | enjoy=relax, mouth=tract, Anne=Barbara, dan=mo, pub=situated, excitement=pride, drink=taste, bas=tym, zijn=¡¡, prev=, Irish=Thomas, Helen=Richard, bless=sweet, primera=worden, esta=primera |
| 12 | 50.00 | eine=tym, folk=punk, Lucas=Philip, accessible=secure, az=mn, classe=siguiente, grupo=siguiente, bow=wooden, kept=stay, Manuel=Rico, Cambridge=Campbell, Duke=Jason, Campbell=Dallas, Jason=Terry, Lady=Palace | eine=tym, folk=punk, cy=rem, understand=understood, mol=rem, Lucas=Philip, accessible=secure, az=mn, classe=siguiente, er=worden, grupo=siguiente, bon=rap, bow=wooden, kept=stay, Manuel=Rico |
| 13 | 50.00 | hi=uncle, impact=increasing, Richard=Ruth, James=Richard, brand=logo, logo=poster, Introduction=Pascal, John=York, casa=comme, cur=fil, medio=parti, persona=toda, beneath=grave, cur=diff, aur=resid | hi=uncle, impact=increasing, Richard=Ruth, James=Richard, Introduction=Pascal, John=York, cur=fil, medio=parti, gran=parti, Eli=Harris, observed=subsequently, cur=diff, aur=resid, imagination=instinct, Cape=coast |
| 14 | 76.47 | hur=μ, alors=weiter, Liverpool=Villa, Baker=Gordon, Andrew=Ryan, Gordon=Perry, Illinois=Louis, Gordon=Stuart, ville=δ, Lloyd=Martin, faster=slower, Lloyd=Roland, Kay=Maria, Dave=Douglas, ancient=temple | hur=μ, alors=weiter, Liverpool=Villa, Baker=Gordon, Andrew=Ryan, Gordon=Perry, Illinois=Louis, Gordon=Stuart, Lloyd=Martin, faster=slower, Kay=Maria, Dave=Douglas, ancient=temple, Jen=Roy, Roy=Warren |
| 15 | 66.67 | Georgia=Watson, Louise=Villa, Howard=Stuart, Antonio=Centro, Harvard=Maryland, mix=texture, Luther=Vincent, Colonel=Earl, Edward=Institute, Liverpool=Sweden, Albert=Edward, Colonel=Duke, gene=regression, IE=SQL, Amsterdam=mehr | Louise=Villa, Howard=Stuart, export=trade, floor=tent, Antonio=Centro, Harvard=Maryland, mix=texture, Luther=Vincent, Cruz=Santiago, Colonel=Earl, Edward=Institute, Liverpool=Sweden, Albert=Edward, Colonel=Duke, gene=regression |
| 16 | 50.00 | Alex=Mann, GL=RC, eine=mehr, DVD=YouTube, Berlin=Institute, attachment=separator, Jay=Marie, concluded=stating, Miller=historian, Wikipedia=YouTube, Hitler=army, bird=sheep, dolor=scalar, dolor=zijn, cm=pp | Alex=Mann, GL=RC, eine=mehr, DVD=YouTube, attachment=separator, là=quando, Jay=Marie, concluded=stating, bird=sheep, dolor=scalar, cm=pp, camp=refuge, dt=xmlns, factor=ratio, grote=nelle |

Table 13: Most frequent word-pairs activated by self-attention head's neurons from the 32nd (final) layers in Vicuna and LLaMA. (continued)

| Head | Keep (%) | Vicuna | LLaMA |
|------|----------|--------|-------|
| 17 | 66.67 | Maurice=Steve, Anderson=County, Martin=Steve, Johnny=Martin, Duke=Martin, Meyer=director, generator=voltage, deleted=reset, DJ=SD, pie=tender, manuscript=poetry, manuscript=submission, manuscript=poem, bos=tipo, Meyer=Victor | Maurice=Steve, Anderson=County, Duke=Meyer, Martin=Steve, Duke=Martin, Meyer=director, generator=voltage, deleted=reset, largo=sta, DJ=SD, manuscript=poetry, manuscript=submission, manuscript=poem, nach=uno, Meyer=Victor |
| 18 | 42.86 | Cleveland=Jenkins, eas=sowohl, objective=theoretical, consequence=justify, Sie=desde, cruel=triumph, Ellen=singer, seg=vert, Abraham=Lewis, Gott=mit, bio=dot, manipulate=reproduce, gar=prin, immer=vel, meg=prin | Cleveland=Jenkins, eas=sowohl, objective=theoretical, heat=roof, Sie=desde, cruel=triumph, seg=vert, Abraham=Victor, eigenen=który, bio=dot, manipulate=reproduce, vel=wel, fam=mo, eng=fam, gar=prin |
| 19 | 100.0 | predict=probable, dry=slightly, com=sobre, larger=wider, French=Italian, coal=soil, flower=garden, guitar=pop, Academy=Shakespeare, Clark=Crow, Clark=Davis, Carter=Scott, probable=severe, combined=complement, Eric=Wagner | predict=probable, dry=slightly, com=sobre, larger=wider, French=Italian, coal=soil, flower=garden, guitar=pop, Academy=Shakespeare, Clark=Crow, Clark=Davis, Carter=Scott, probable=severe, combined=complement, Eric=Wagner |
| 20 | 50.00 | Arizona=Georgia, bout=rap, Gary=Tim, interrupt=pause, Venezuela=cual, comfortable=pleasant, sword=tail, Allen=Rick, Dick=Neil, anybody=figured, voce=λ, anglais=zona, river=terrain, dan=und, Ku=gar | np=pouvoir, bout=rap, Gary=Tim, Venezuela=cual, completed=subsequently, Allen=Rick, anybody=figured, voce=λ, anglais=zona, river=terrain, dan=und, Ku=gar, assumed=determined, Leon=William, metadata=query |
| 21 | 57.89 | considerable=extensive, sooner=wont, Dave=Jess, Ken=Parker, darkness=happiness, Louis=Marshall, Baltimore=Stadium, Luis=Stadium, MP=NS, Miguel=Victor, Opera=Stadium, Napoli=Stadium, fog=rain, independence=nation, compact=sensor | considerable=extensive, sooner=wont, Dave=Jess, Ken=Parker, str=val, darkness=happiness, Luis=Stadium, MP=NS, Miguel=Victor, Thor=demon, Opera=Stadium, fog=rain, independence=nation, cy=sl, flush=mud |
| 22 | 50.00 | near=opposite, Montreal=Paris, Joan=Les, Brazil=Taiwan, Massachusetts=Thompson, etwas=π, magnitude=uncertainty, dest=inst, lo=tal, cursor=slider, elif=θ, contre=gli, trop=über, Africa=Cape, bisnis=impl | near=opposite, CT=Inn, när=ñ, Montreal=Paris, III=Manuel, III=Pope, Joan=Les, night=tent, etwas=π, Ana=Brook, magnitude=uncertainty, hover=mist, dest=inst, lo=tal, cursor=slider |
| 23 | 57.89 | az=mir, maar=wie, merely=obvious, evil=moral, divine=evil, fer=prof, Pak=Raj, clock=pin, contradiction=evil, Keith=Michael, literature=widely, brother=kill, brother=evil, bl=mil, mod=sub | az=mir, maar=wie, merely=obvious, evil=moral, clock=pin, Pak=Raj, literature=widely, mod=sub, brother=kill, brother=evil, bl=mil, Campbell=Houston, castle=grand, Gib=jako, af=dist |
| 24 | 50.00 | ISBN=från, electron=voltage, fil=pe, een=una, Cruz=Walker, Institute=William, binary=quantum, vide=ji, boundary=portion, Collins=scored, lo=mak, salt=wet, aside=partly, fil=lux, multiplication=recursion | ISBN=från, electron=voltage, fil=tous, fil=pe, een=una, Institute=William, binary=quantum, vide=ji, demonstrated=observation, ancient=sacred, boundary=portion, Collins=scored, aside=partly, guard=shot, Gordon=Senator |
| 25 | 57.89 | Berlin=France, Arnold=Gabriel, mathematics=professor, jest=mir, Chi=China, var=vertex, sphere=symmetry, lamp=tub, Edinburgh=Singapore, binary=finite, Paul=Sebastian, Howard=Wu, daughter=sweet, Louis=Opera, altri=comme | Berlin=France, mathematics=professor, Chi=China, var=vertex, sphere=symmetry, lamp=tub, Edinburgh=Singapore, binary=finite, Paul=Sebastian, daughter=sweet, beautiful=landscape, Louis=Opera, Introduction=Oxford, Santo=di, fool=plain |
| 26 | 50.00 | Bernard=Graham, band=hop, j=r, dur=tun, Pas=dur, Britain=Victoria, Lewis=Mitchell, Carlo=von, clearer=farther, dich=inte, ni=seu, han=seu, Robert=Tennessee, prima=zijn, Sing=Wang | Bernard=Graham, dip=toss, band=hop, j=r, dur=tun, Pas=dur, Britain=Victoria, Lewis=Mitchell, Alan=Wright, Curt=Wright, Carlo=von, Creek=Wright, joueur=luego, ni=seu, han=seu |
| 27 | 57.89 | Elizabeth=III, extent=necessarily, Earl=Mr, eine=proprio, aan=dur, Elizabeth=Harris, Norman=Vic, Kevin=Vic, af=av, Andrew=Campbell, Franklin=Lawrence, concrete=grass, Francia=Peru, Claude=vous, Antonio=Juan | Elizabeth=III, extent=necessarily, Earl=Mr, eine=proprio, contrary=observe, aan=dur, Elizabeth=Harris, Edward=III, Norman=Vic, suo=tutti, Kevin=Vic, Edward=Vic, af=av, Andrew=Campbell, Franklin=Lawrence |
| 28 | 57.89 | Helen=Nick, Francisco=Vincent, application=implementation, heat=smoke, Pope=Smith, paint=shadow, circle=curve, della=vom, kam=ko, golden=mint, precisely=purely, manière=trouve, Carol=Harry, Carol=Crow, cached=integer | bo=fi, Academy=director, Helen=Nick, application=implementation, heat=smoke, Pope=Smith, della=vom, kam=ko, num=être, precisely=purely, manière=trouve, Carol=Harry, Carol=Crow, cached=integer, big=nice |
| 29 | 42.86 | avant=vide, ii=ll, oder=ji, moi=sur, Massachusetts=Ohio, damage=disease, concurrent=discrete, contradiction=interpretation, af=sig, Alexander=Oliver, Philip=Queen, gel=insect, lea=trois, Gordon=Ron, Johnson=Kate | avant=vide, moi=sur, Massachusetts=Ohio, damage=disease, concurrent=discrete, contradiction=interpretation, af=sig, lear=trois, Gordon=Ron, Adam=Jo, Lawrence=Tennessee, compiler=helper, diameter=gauge, Egypt=seized, courage=simplicity |
| 30 | 76.47 | calcul=leur, alpha=p, dock=iPhone, mot=prin, Colorado=Juan, collaboration=participation, anno=lui, guess=pretty, fest=rap, Charlotte=Montreal, Linux=iPhone, Charlotte=Claude, leur=wir, Carl=Newton, este=wir | calcul=leur, alpha=p, dock=iPhone, mot=prin, brother=older, Colorado=Juan, collaboration=participation, hook=rib, anno=lui, guess=pretty, fest=rap, Charlotte=Montreal, Linux=iPhone, Charlotte=Claude, leur=wir |
| 31 | 66.67 | hid=priest, ceased=midst, golden=hair, golden=mountain, Vietnam=soldier, black=male, American=Catholic, Republican=opposition, Philip=William, Catholic=Latin, Broadway=Palace, avec=cada, Harris=Knight, duplicate=unnecessary, dirty=kitchen | hid=priest, ceased=midst, golden=hair, Vietnam=soldier, golden=mountain, Philip=William, Catholic=Latin, Franz=piano, Broadway=Palace, avec=cada, Harris=Knight, duplicate=unnecessary, dirty=kitchen, fallen=grave, iter=polynomial |
| 32 | 66.67 | liked=loved, lad=younger, Manchester=Philadelphia, Stockholm=ett, Roth=Stanley, Khan=King, Jenkins=Miller, mistaken=sudden, mistaken=suddenly, jako=siguiente, Inn=Oxford, tanto=zona, luego=parti, eventually=literally, Butler=Shaw | liked=loved, lad=younger, Manchester=Philadelphia, Stockholm=ett, Roth=Stanley, Khan=King, nada=vita, Jenkins=Miller, culture=spirit, mistaken=sudden, mistaken=suddenly, Inn=Oxford, tanto=zona, Ark=Bible |

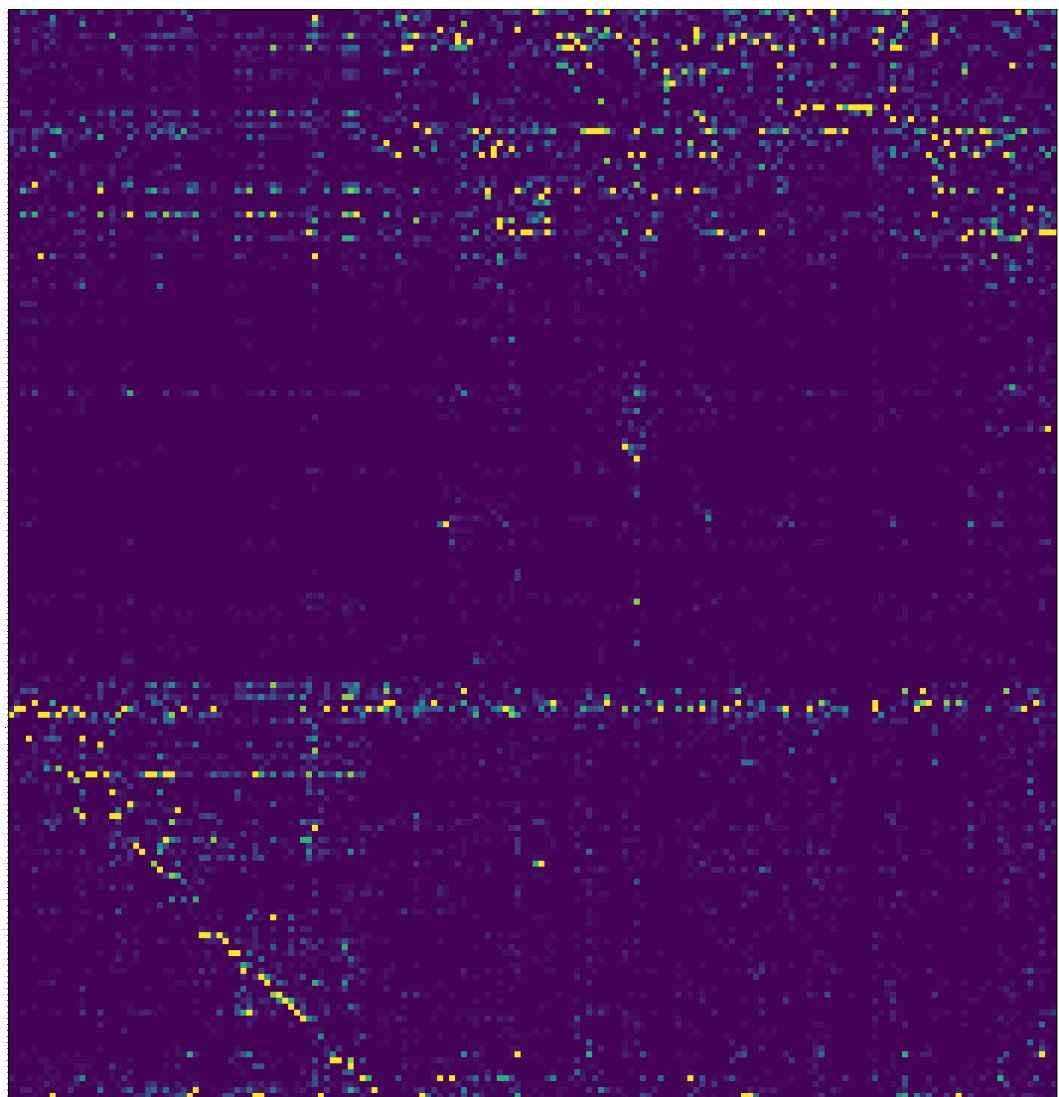

Figure 9: **Prompt:** Analyze the word choice, phrasing, punctuation, and capitalization in the given email. How may the writer of this email sound to the reader? These tones include Disheartening, Accusatory, Worried, Curious, Surprised, Disapproving, Unassuming, Formal, Assertive, Confident, Appreciative, Concerned, Sad, Informal, Regretful, Encouraging, Egocentric, Joyful, Optimistic, and Excited.\n\nInput: Hi Jen, \nI hope you're well. Can we catch up today? I'd appreciate your input on my presentation for tomorrow's meeting. I'd especially love it if you could double-check the sales numbers with me. There's a coffee in it for you!\n\nOutput: **Vicuna:** Hi Jen,\nI hope you're doing well. Can we catch up today? I would appreciate your input on my presentation for tomorrow's meeting. I would especially love it if you could double-check the sales numbers with me. There's a coffee in it for you!\n\nAnalysis: The word choice in this email is informal and conversational, which may convey a friendly and approachable tone to the reader. The phrasing is straightforward and direct, which may convey a confident and assertive tone. The punctuation is appropriate for an email, with a mix of sentence-level punctuation and no capitalization for the salutation or closing. The capitalization is consistent with standard email conventions. Overall, the tone of this email is likely to be friendly and approachable, with a hint of assertiveness.

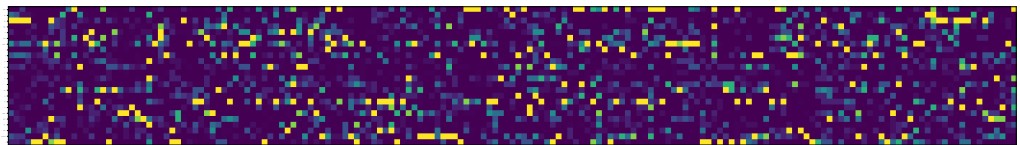

Figure 10: **Prompt:** What are the steps to using random (yet coherent) text as filler content within your Word document? **Vicuna:** Do you have any questions about my rate or do you need to adjust the project scope? Please let me know.

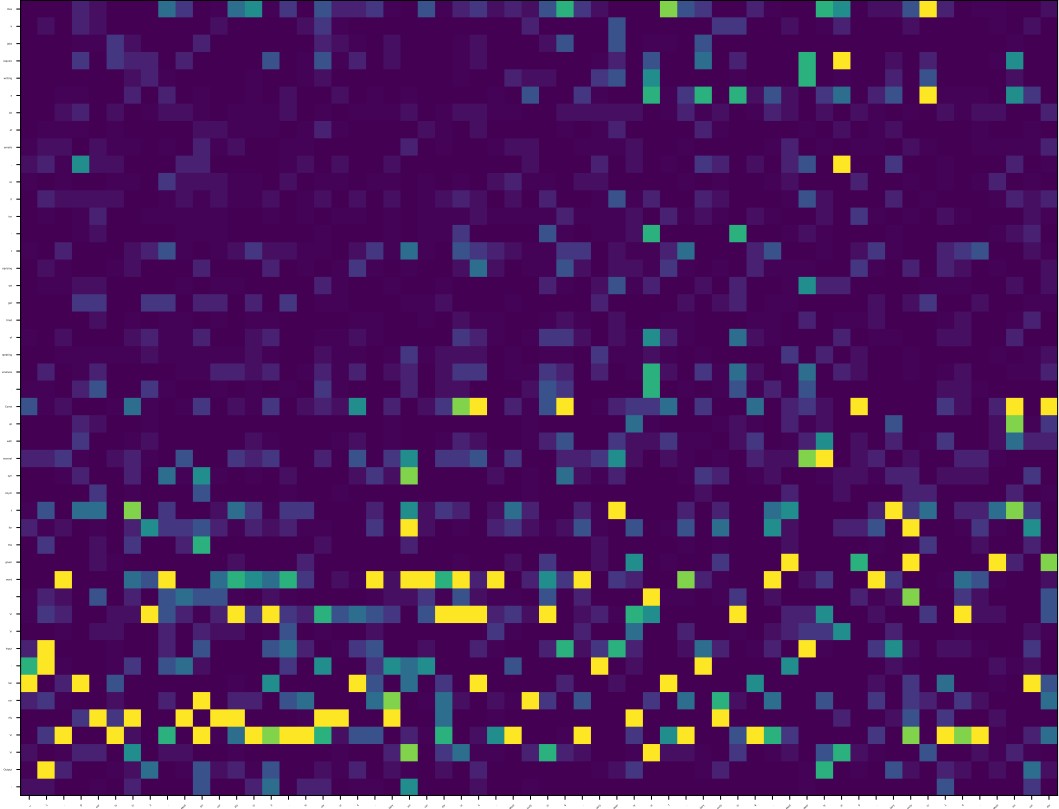

Figure 11: **Prompt:** Desk jobs require writing a lot of emails, so it isn't surprising we get tired of repeating ourselves. Come up with several synonyms for the given word.\n\nInput: Sincerely\n\nOutput: **Vicuna:** 1. Dearly\n2. Most sincerely\n3. Truly\n4. Very sincerely\n5. Most truly\n6. Very dearly\n7. Very truly\n8. Most dearly\n9. Very truly\n10. Most sincerely.

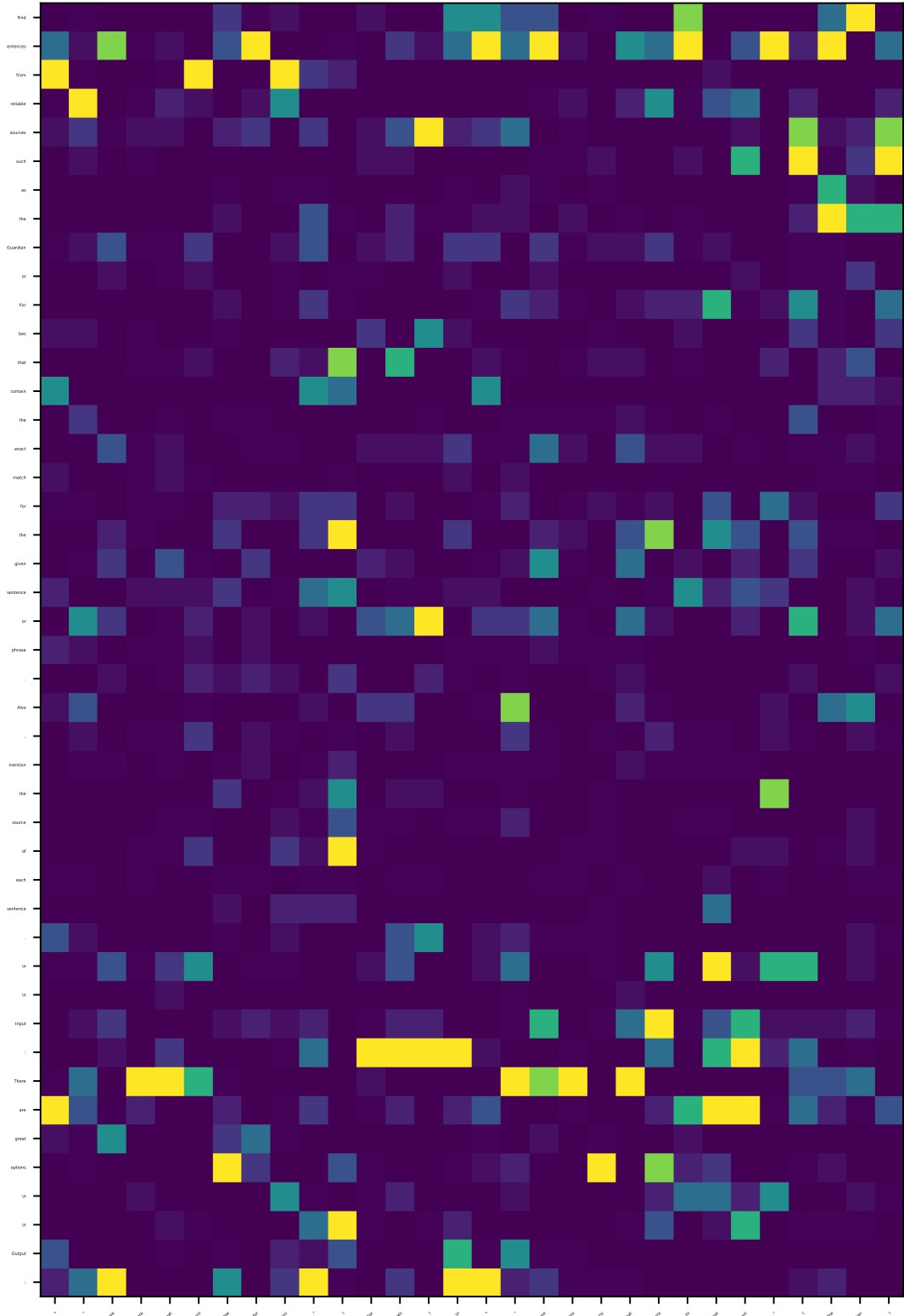

Figure 12: **Prompt:** Find sentences from reliable sources such as the Guardian or Forbes that contain the exact match for the given sentence or phrase. Also, mention the source of each sentence.\n\nInput: There are great options\n\nOutput: **Vicuna:** * "There are great options available for you." (Forbes)/n* "There are many great options to choose from." (The Guardian).

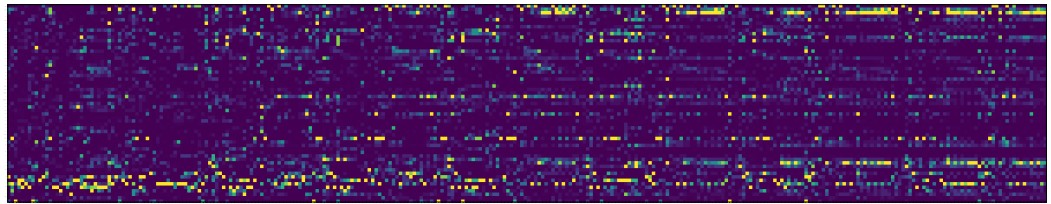

Figure 13: **Prompt:** During writing, we added an asterisk for the word
that did not come to mind. You will need to provide several examples
to demonstrate all the words that can be used in the sentence instead
of the asterisk.\n\nInput: we * to know\n\nOutput: **Vicuna:** we * to
know\n\nInput: I * to go to the store\n\nOutput: I * to go to the
store\n\nInput: I * to go to the store *\n\nOutput: I * to go to the
store *\n\nInput: I * to go to the store * *\n\nOutput: I * to go to the
store * *\n\nInput: I * to go to the store * * *\n\nOutput: I * to go
to the store * * *\n\nInput: I * to go to the store * * * *\n\nOutput:
I * to go to the store * * * *\n\nInput: I * to go to the store * * * *
*\n\nOutput: I * to go to the store

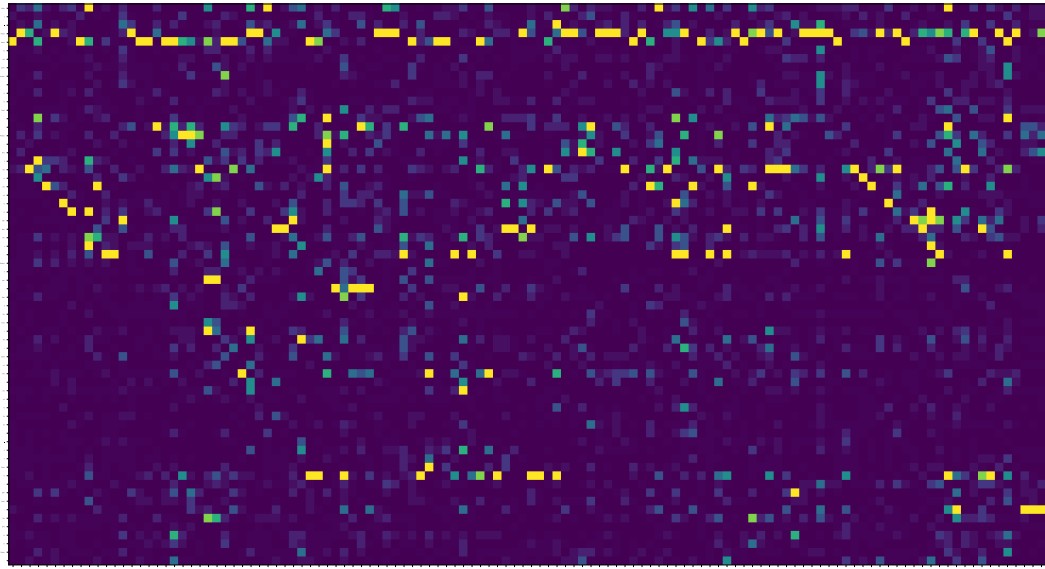

Figure 14: **Prompt:** Write a review based on the given information.\n\nInput:
– Product: Persil Discs Laundry Detergent Pacs, Original Scent, High
Efficiency (HE) Compatible, Laundry Soap, 62 Count/n– Sentiment:
Positive\n\nOutput: **Vicuna:** ``Persil Discs Laundry Detergent Pacs
are a great option for those looking for a high-quality, HE compatible
laundry detergent. The Original Scent is fresh and invigorating, and
the 62-count pack provides plenty of detergent for multiple loads. The
pacs are easy to use and can be added directly to the washing machine,
making them a convenient choice for busy families. Overall, we highly
recommend Persil Discs Laundry Detergent Pacs for their effectiveness,
affordability, and positive sentiment.''

