# OpenReview forum: "From Language Modeling to Instruction Following: Understanding the Behavior Shift in LLMs after Instruction Tuning"
_ICLR.cc/2024/Conference — ICLR 2024 Conference Withdrawn Submission_

### Official Review · Reviewer_zeP7 · 2023-10-29

**Soundness:** 2 fair
**Presentation:** 1 poor
**Contribution:** 3 good
**Rating:** 5
**Confidence:** 4

**Summary:**

This paper does a comprehensive comparison between a pre-trained model (LLaMa) and two instruction fine-tuned models (Vicuna and Alpaca). The authors look at the importance of particular types of input tokens (like instruction words) for the gold output probabilities, as well as what parts of the inputs models focus on. They find that instruction-tuned models suffer less from the known problem that models focus more on the first and last part of the prompt than the middle, and that when Vicuna follows user instructions, instruction words are more important. The authors also use an interpretability technique to look at concepts the FFN and the attention heads encode. finding, among other things, that there is a shift in represented concepts after fine-tuning (more focus on math, writing, coding), but no shift in linguistic concepts.

**Strengths:**

This paper does a lot of experiments and seemingly has some very interesting results that can contribute to our understanding of the behavior shift after instruction tuning:
- there is a shift in representation of concepts like math and coding but not in linguistic concepts after instruction fine-tuning
- instruction-tuned models have less of a problem with not focusing on the middle of the prompt, but the problem is still there

**Weaknesses:**

The paper is not written very clearly, and is very hard to follow because of it. There are many errors and difficult-to-interpret formulations. Additionally, the authors sometimes compare to the baseline model LLaMa, and sometimes do not. The latter results are difficult if not impossible to interpret. The qualitative results are sometimes based on a single prompt or only a small subset of analyses concepts, which makes it hard to know if the results hold for other prompts and for the other concepts. In some more detail:

- Section 3.1.3 is seemingly based on a single prompt. The results here are also very hard to interpret from just this single saliency plot. I'd say you can leave this out entirely and just pose this as a hypothesis and confirm in section 3.2.3, or alternatively somehow show that it holds for more than just this single prompt.
- You only show the importance density on instruction words for Vicuna in Table 1. How can we interpret the shift between LLaMa and Vicuna here if the importance density for LLaMa is not mentioned? You say *"Table 1 underscores that instruction fine-tuned models excel in identifying and harnessing instruction words from user prompts for response generation"*, but how can you say that without a baseline comparison? Maybe LLaMa will follow instructions more when more importance is on instruction words as well?
- In table 2 you mention 5 of 300 principal component concepts. The reader is left wondering if the rest is not interpretable? And how does this compare to LLaMa? Similarly for observation 5, you talk about these 4 concepts, and it's unclear what's going on with the others or how many there even are.
- You never mention in the main text that you use chatGPT to identify concepts.
- In the discussion you claim that your results show the importance of prompt diversity, but it's unclear to me how they show that.

As mentioned, the text is very difficult to follow:
- After reading the abstract and introduction, I have very little understanding of the findings. For example, you keep referring to the "lost-in-the-middle" issue without saying what this is. After reading the entire paper, it's clear that this can be explained in very few words.
- You don't mention you compare to a non-fine-tuned baseline until section 3
- In the introduction you say that instruction-tuning updates model weights different from pre-training, and it's unclear what you mean since pre-training of course also updates weights.
- In the introduction you use an example-sentence "me and him goes to a stores for buying a apple" that has many errors, and it seems like this is on purpose to highlight something about the analysis, but it's completely unclear to the reader what this is supposed to highlight.
- Many examples of difficult to follow sentences, e.g. : Last paragraph on page 1; "What the encoded knowledge evolves after instruction fine-tuning?";  you write LLaMa differently many times (LLama, LLaMa, LLaMA);  "differed to typical language models", end of page 5; "one of the nice property by using l1/lp density function is if two input tokens ... would receives more .."; "to verify our assumption that a model is following instruction if it identify and use instruction words" should be "identifies and uses"; "where each prompt was manually annotated the instruction part"; "Before applying our method, we crease a new vocabulary"; "the build-in LLaMA vocabulary"; "indicating multi-linguistic knowledge is forgetting" should be "forgotten" or something; etc.

To summarise;

I think this paper has some very interesting results that merit publication, however the current write-up is very difficult to follow and the reader is left with many questions. Most importantly; some observations seem based on only a very small part of the analyses, the authors do not always compare to the baseline LLaMa, and most of the paper is difficult to follow.

**Questions:**

- Why do you include section 2.1? That is very common notation and the Transformer architecture can also be assumed known I'd say.
- Why do you use a different set of hyper-parameters for 3.2.2 than for 3.2.1 when you're trying to verify the assumption in the latter with the former?
- Could you add some justification for the analysis in 4.1? Why is this a sensible thing to do?
- Could you justify the linear approximation of the importance and especially why it's OK to just incorporate the embedding of the input token?

---

> ### Author Response · Authors · 2023-11-15
> **Response to Reviewer zeP7 - Part 1/2**
>
> Thanks for your comments, we have prepared the following responses to address your concerns.
>
> **W-1: Section 3.1.3 is seemingly based on a single prompt.**
>
> We design the experiments in Section 3.1 as a qualitative analysis. Although we discuss our observation via one single instance in the main paper, we actually check that it holds for more cases, and we provide their saliency maps in Appendix A.2. We quantify this observation in Section 3.2 by comparing the density of importance scores on instruction words over the followed and un-followed instances. We observe a higher density score on the followed instances. Thus, both visualization (Section 3.1) and quantitative experiments (Section 3.2) demonstrate that instruction words constantly guide the generation process as the key behavior of LLMs to align with human preference. In our revised version, we will emphasize that results in Section 3.1.3 pose a hypothesis and will be confirmed in Section 3.2.
>
> **W-2: Expecting results of importance density from LLaMA.**
>
> This experiment aims to verify our hypothesis drawn from Section 3.1.3 quantitatively. Thus, the revised version would rephrase this conclusion with a more proper expression that "Table 1 underscores the models generate responses more aligned with human preference by better identifying and harnessing instruction words from user prompts." In addition, we believe it is a good suggestion to compare LLaMA and Vicuna under this setting. Thus, we are conducting the experiment with LLaMA. We will post the results and the rephrased conclusion in our revised version, which will be uploaded soon later.
>
> **W-3.1: Expecting more interpretation cases in Table 2.**
>
> Table 2 only presents the interpretation of 5 principal components because we cannot include those cases that are hard to present in the main paper within such limited spaces. However, in Appendix C.2 and Figure 7, we demonstrate that over 80% of principal components from the last layer of Vicuna/LLaMA are interpretable by ChatGPT. In addition, Table 9 and Table 10 include more interpretable principal components from both the last and the first layers. Overall, the word lists always show clear and concise concepts to humans. We will release the full annotated dataset once the paper is accepted.
>
> **W-3.2: Table 4 only compares the models over 4 concepts.**
>
> The categories in Table 4 are *not* interpreted concepts. They are 4 scenarios and 4 linguistic disciplines used to study the shift of the interpreted concepts after instruction-tuning.
>
> **W-4: The main text should mention that ChatGPT is used to identify concepts.**
>
> Thanks for pointing this out! The revised version will emphasize in the main text that ChatGPT is considered as the machine annotator to scale up our interpretation method.
>
> **W-5: It is unclear how the results show the importance of prompt diversity.**
>
> Finding 1 provides evidence that a model is more helpful for humans if it can better identify those instruction words. In addition, Finding 3 demonstrates that self-attention heads would directly learn the word-word relations with instruction words (verbs). Since diverse prompts would offer diverse instruction words, we could expect the model to achieve a better performance by training on diverse prompts.
>
> __W-6/W-7: The abstract and introduction could be improved.__
>
> Thanks for all these suggestions, such as explaining terms earlier and introducing the research methods more clearly. Our revised version will take them into consideration.
>
> __W-8: In the introduction, you say that instruction-tuning updates model weights differently from pre-training, and it's unclear what you mean since pre-training of course, also updates weights.__
>
> The original writing wants to point out that although related works have studied _the weights of pre-trained models_, their findings could not extend to instruction-tuned models since instruction tuning has the chance to modify the pre-trained weights.
>
> __W-9: The example sentence in the introduction is not clear enough to the readers.__
>
> We want to provide a prompt such as "fix grammar: me and him goes to an stores for buy an apples" to demonstrate the _instruction_ and _context_ parts. In this case, "fix grammar:" is the instruction part, while the rest is the context part. The revised version will describe this case more accurately with more sentences.
>
> __W-10: Typos and sentences difficult to be followed.__
>
> Thanks for pointing these sentences out! Our revised version would try to improve these sentences.

---

> ### Author Response · Authors · 2023-11-15
> **Response to Reviewer zeP7 - Part 2/2**
>
> __Q-1: Why do you include Section 2.1? That is very common notation and the Transformer architecture can also be assumed known I'd say.__
>
> We include Section 2.1 to provide formal guidance of notations in this work so that the readers could easily follow the methods proposed in different sections. In our revised version, we will further refine this section, and introduce some observations that motivate this study from prior works, such as the "lost-in-the-middle" phenomenon.
>
> __Q-2: Why do you use a different set of hyper-parameters for 3.2.2 than for 3.2.1 when you're trying to verify the assumption in the latter with the former?__
>
> In our study, L=10 is set as a constant for all experiments. $b$ is a hyper-parameter to consider how weak the importance score could be considered as noise. We intentively set $b$ to 0 because our aim was to present all available information, even if it included some noise, to our readers through heatmap visualizations. We were concerned that the readers might raise objections if we did not provide a comprehensive overview of the salient map, hence the choice of $b=0$.
>
> **Q-3: Explain why it is sensible to interpret feedforward networks with PCA.**
>
> Section 4 aims to interpret the knowledge encoded in feedforward networks; the most straightforward way is interpreting their neurons (vectors of the weight matrixes). However, many studies have pointed out that neurons are polysemantic [1,2,3,4], leading to a poor interpretability to humans. Therefore, we propose to decompose these neurons (patterns) via principal component analysis and understand the basis of these neurons.
>
> > [1] Trenton Bricken, et. al. Towards Monosemanticity: Decomposing Language Models With Dictionary Learning. _Anthropic Blog_, 2023.
> >
> > [2] Arora, Sanjeev, et al. "Linear algebraic structure of word senses, with applications to polysemy." *Transactions of the Association for Computational Linguistics* 6 (2018): 483-495.
> >
> > [3] Elhage, Nelson, et al. "Toy models of superposition." *arXiv preprint arXiv:2209.10652* (2022).
> >
> > [4] Cunningham, Hoagy, et al. "Sparse autoencoders find highly interpretable features in language models." *arXiv preprint arXiv:2309.08600* (2023).
>
> __Q-4: Could you justify the linear approximation of the importance and especially why it's OK to just incorporate the embedding of the input token?__
>
> We want to prove that why equation $I_{n,m}=p(y_m|Z_m) - p(y_m|Z_{m,/n})$ could be approximated with the first first-order Tylor extension to $I_{n,m}\approx \frac{\partial f(y_m|z_m)}{\partial \mathbf{E}_i[x_n]} \cdot \mathbf{E}_i[x_n]^\top$.
>
> We rewrite $p(y_m|Z_m)$ as $f(y_m|\mathbf{Z}_m)$, where $f$ is the language model, $\mathbf{Z}_m\in\mathbb{R}^{(N+m-1)\times d}$ are the word embeddings of the input token sequence $Z_m$, and the $d$-dimensional word embeddings of a token $w\in Z\_m$ is defined as $\mathbf{E}\_i[w]$.
>
> Thus, we have $I\_{n,m}=f(y\_m|\mathbf{Z}\_m) - f(y\_m|\mathbf{Z}\_{m,/n})$, where we let the $n$-th row vector of \$\mathbf{Z}_{m,/n}\$ be zeros.
>
> The first order Taylor expansion of $f(y\_m|\mathbf{Z}\_m)$ around $\mathbf{Z}\_{m,/n}$ is
> \$\$
> f(y\_m|\mathbf{Z}\_m)\approx f(y\_m|\mathbf{Z}\_{m,/n})+\frac{\partial f(y\_m|\mathbf{Z}\_m)}{\partial\mathbf{Z}\_m}\Bigg|\_{\mathbf{Z}\_{m,/n}}\cdot (\mathbf{Z}\_m-\mathbf{Z}\_{m,/n})^\top.
> \$\$
>
> Since $\mathbf{Z}_{m,/n}$ differs from $\mathbf{Z}_m$ only in the $n$-th row, the term $\mathbf{Z}\_m-\mathbf{Z}\_{m,/n}$ is just the vector $\mathbf{E}_i[x_n]$ and the other rows are zero.
>
> Therefore, the above equation could be simplified as:
> \$\$
> f(y\_m|\mathbf{Z}\_m)\approx f(y\_m|\mathbf{Z}\_{m,/n})+\frac{\partial f(y\_m|\mathbf{Z}\_m)}{\partial\mathbf{E}\_i[x_n]}\cdot \mathbf{E}\_i[x\_n]^\top.
> \$\$
> Bring this approximation back to the definition of $I_{n,m}$, we have $I_{n,m}\approx \frac{\partial f(y_m|\mathbf{Z}_m)}{\partial\mathbf{E}_i[x_n]}\cdot \mathbf{E}_i[x_n]^\top$.
>
> The above derivation justifies the approximation of the importance with gradients over the word embedding of the input token $x_n$.

---

> ### Comment · Reviewer_zeP7 · 2023-11-20
> **Thanks for the rebuttal**
>
> Thanks for giving answer to my questions, for Q1-3 the answers are clear. With this message I want to respond to your answers to some of my weaknesses.
>
> Even if you clarify that Section Section 3.1.3 is just a hypothesis, the plot you use is still very hard to interpret.
>
> For my second point about the phrase *"Table 1 underscores that instruction fine-tuned models excel in identifying and harnessing instruction words from user prompts for response generation"* you say you will add results, but for now this is still an unsupported claim in the paper.
>
> The fact that you cannot discuss all principal components of table 2 in the main text is understandable, but the reader is now left questioning this without fully going through the appendix. I think even with limited space you should have a summary of the findings presented in the appendix for these kinds of things.
>
> Additionally, I still don't understand after your explanation how the results show the importance of prompt diversity. It seems like you're saying that instruction words help a model in performing tasks, and therefore prompt diversity is important. I don't see how this is related to diversity. You say *"Since diverse prompts would offer diverse instruction words, we could expect the model to achieve a better performance by training on diverse prompts."*, but I don't believe you can claim this without actually doing that experiment.
>
> Unfortunately, because of the above, I cannot change my current rating. I believe this manuscript needs too much rewriting to be published.

---

> > ### Author Response · Authors · 2023-11-21
> > **Response to additional feedback from Reviewer zeP7**
> >
> > Thanks for sharing your additional concerns about our work. Our revised version has addressed your concerns.
> >
> > __Concern-1: Even if you clarify that Section Section 3.1.3 is just a hypothesis, the plot you use is still very hard to interpret.__
> >
> > Our revised version has updated the cases plotted in Section 3.1.3. In the initial submission, we drew the salient map with the importance scores with the golden response. However, in the revision submission, the responses were changed to model-generated ones. In the revision, the differences in the salient maps between the Vicuna and LLaMA could be clearly identified.
> >
> >
> > __Concern-2: For my second point about the phrase "Table 1 underscores that instruction fine-tuned models excel in identifying and harnessing instruction words from user prompts for response generation" you say you will add results, but for now this is still an unsupported claim in the paper.__
> >
> > Our revised version has included the importance density from LLaMA in Appendix B.1.1 and rephrased this claim into "In summary, Table 1 underscores the strong correlation between the importance density scores of instruction words and the instruction following capability. In Appendix B.1.1, we show that Vicuna achieves greater importance density scores on instruction words compared to LLaMA across the three datasets, indicating instruction tuning empowers the pre-trained model in better identifying and harnessing instruction words from user prompts for response generation."
> >
> >
> > __Concern-3: I think even with limited space you should have a summary of the findings presented in the appendix for these kinds of things.__
> >
> > Our revised version has included a summary of the findings in Obs-4 as follows: "Limited by the limited space of the main text, we select 5 representative principal components and their explanations from the last feed-forward network of Vicuna and display them in Table 2. However, Appendix D.2 shows that around 60% of the first 300 principal components from the middle layers of Vicuna could be interpreted by ChatGPT, and further reports the statistical results of the word frequency from the machine-annotated descriptions. More interpretation cases are available in Table 10 and Table 11."
> >
> >
> > __Concern-4: I still don't understand after your explanation how the results show the importance of prompt diversity.__
> >
> > Finding 1 demonstrates the importance of instruction words for generating responses aligning with human intentions. Finding 3 demonstrates that the self-attention heads (from the bottom and middle layers) explicitly learn word-word relations with instruction verbs. With these two findings, we could derive that presenting different instruction words during SFT is beneficial to the instruction-following capability. Therefore, we made the claim that "Since diverse prompts would offer diverse instruction words, we could expect the model to achieve a better performance by training on diverse prompts." We want to point out that this claim has been empirically proved by LIMA [1], and this research provides an inherent perspective to demonstrate their finding.
> > > [1] Zhou, Chunting, et al. "Lima: Less is more for alignment." arXiv preprint arXiv:2305.11206 (2023).

---

> > > ### Comment · Reviewer_zeP7 · 2023-11-22
> > > **Thanks for the responses**
> > >
> > > Thank you for your responses and revisions. Indeed the plot is much clearer now, and the text in 4.1.3 as well. However, I remain with my original point; I believe this paper needs another cycle of reviews because of the state of the initial manuscript and that of the revision.

---

### Official Review · Reviewer_QqaN · 2023-10-30

**Soundness:** 2 fair
**Presentation:** 2 fair
**Contribution:** 2 fair
**Rating:** 6
**Confidence:** 3

**Summary:**

This paper investigates the behavior shift of LLMs after the instruction-following finetuning step. To this end, it introduces several explanation methods enabling the study of the effect of the finetuning procedure on the pretrained model. With a series of examples, the paper shows that:
- finetuned LLMs recognize better the instruction than the pretrained counterpart
- the finetuning process aligns the kowledge in FF layers with user tasks
- self-attention heads learn better word-word relations with instruction verbs after the finetuning process.

**Strengths:**

This paper develops a set of local and global methods to study large language models. These methods are more general than the scope of this paper and can be used by the literature to analyze any LM for any given task.

Also, the paper introduces good notations that help to understand the different methods and experiments.

**Weaknesses:**

- Obs-1 in 3.1.3 ("instruction fine-tuning helps the models distinguish between instruction and context words more accurately") is not convincing. That is, it is not clear how Figure 1 shows that instruction finetuning helps the model distinguish between instruction and context words more accurately. For instance, I cannot see that the instruction part is brighter than the context one in the figure, or that the finetuned model focuses more on instructions compared to the pretrained one. Some quantitative metrics could be helpful to interpret this figure.

- The introduced metrics rely on a few introduced parameters, namely L, b, p. The choice of these parameters was not justified in the paper and it does change depending on the paragraph. Some justification for the choices needs to be added to the paper.

**Questions:**

- I don't understand the derivation of S_{n,m} from I_{n,m} and its intuition. Can you add a few sentences about this metric and its need compared to using directly I_{n,m}? Also, how do the added hyper-parameters L and b need to be chosen?

- I find the results in Table 1 interesting. However, can you also add the results of importance density S_{n,m} for the pretrained model in the same task? This will give a good baseline to assess the impact of the finetuning process. As it is, it is not a study of the effect of the finetuning step

Misc/typos:
- "Also, what the encoded knowledge evolves after instruction fine-tuning?” --> “Also, how the encoded knowledge evolves after instruction fine-tuning?”
- I am not sure what this sentence means: we collect an annotated dataset, where each prompt was manually annotated the instruction part
- Hugginface library → Huggingface library
- To verify our assumption that a model is following instruction if it identify and use instruction words to guide the generation, we collect an annotated dataset --> “To verify our assumption that a model is following instruction*s* if it identif*ies* and use*s* instruction words to guide the generation, we collect an annotated dataset”
- Thus, it is naturally to represent this vector with the words having largest projection length → “it is natural”
- Before applying our method, we crease a new → we create

---

> ### Author Response · Authors · 2023-11-15
> **Response to Reviewer QqaN - Part 1/1**
>
> Thanks for your comments, we have prepared the following responses to address your concerns.
>
> **W-1: Section 3.1.3 is not convincing since it is made by analyzing the visualization of an instance without quantitative analysis.**
>
> We design the experiments in Section 3.1 as a qualitative analysis. Although we discuss our observation with one single instance in the main paper, we confirm it holds for more cases and provide their saliency maps in Appendix A.2. We then quantify this observation in Section 3.2. Thus, both visualization (Section 3.1) and quantitative experiments (Section 3.2) demonstrate that instruction words constantly guide the generation process as the key behavior of LLMs to generate responses aligned with humans. The revised version will emphasize the connection between results in Section 3.1 and Section 3.2.
>
> **W-2: Settings of hyper-parameters.**
>
> In our study, L=10 is set as a constant for all experiments. $b$ is a hyper-parameter to consider how weak the importance score could be considered as noise. We intentively set $b$ to 0 because our aim was to present all available information, even if it included some noise, to our readers through heatmap visualizations. We were concerned that the readers might raise objections if we did not provide a comprehensive overview of the salient map, hence the choice of $b=0$. For the quantitative analysis in Section 3.2, we used a grid search on $b$ and $p$ to minimize the p-values.
>
> **Q-1: I don't understand the derivation of S_{n,m} from I_{n,m} and its intuition.**
>
> We derive $I_{n,m}$ to $S_{n,m}$ with two intuitions. Firstly, we consider that $I\_{n,m\_1}$ and  $I\_{n,m\_2}$ are not directly comparable since their gradients are computed starting from different confidence scores, i.e. $f(y\_{m\_1}|Z\_{m\_1})$ and $f(y\_{m\_2}|Z\_{m\_2})$. Predicting token m_1 and token m_2 are two word classification tasks with different target label sets. For example, you want to generate the word sequence "this is a short sentence", and let n=this, m_1 = is and m_2 = short. Since "this is" is a very common phrase, while the word "short" is less common in following the word "a", we have $f(y\_{m\_1}|Z\_{m\_1})<f(y\_{m\_2}|Z\_{m\_2})$. To make them comparable across each prediction word, we normalize them from 1 to L as $\left\lceil L \times \frac{I_{n,m}}{\max\_{n^\prime=1}^N I_{n^\prime,m}}\right\rceil$. Secondly, we consider that estimating the gradients over such a large language model may introduce noise. Thus, we sparsify the importance by setting that importance lower than b as 0.
>
> **Q-2: Adding the results of importance density $S_{n,m}$ for LLaMA.**
>
> Yes, we are conducting the experiment and manually labeling follow/unfollowed for the outputs from LLaMA. The revised version will bring the result with LLaMA.
>
> **Misc/typos.**
>
> Thanks a lot for pointing out these typos. We will fix them in our revised version. We will also conduct another self-check for those typos.

---

### Official Review · Reviewer_e43k · 2023-10-31

**Soundness:** 2 fair
**Presentation:** 2 fair
**Contribution:** 2 fair
**Rating:** 3
**Confidence:** 4

**Summary:**

The paper presents a suite of methods designed at understanding and interpreting token importance for LLMs. The authors develop a series of different methods, which build on each other and measure importance between input and output tokens; between input tokens and the entire output sequence; and the "concept-level" knowledge in feedforward networks. They evaluate these metrics in the context of LLaMA and its instruction-tuned variants Vicuna, Alpaca, and connduct this analysis over the Self-Instruct, LIMA, and MT-Bench datasets. The authors highlight 3 key findings: (1) fine-tuning enables LMs to recognize "instruction words" from prompts; instruction-tuning "aligns knowledge in feed-forward layers with user-oriented tasks with minimal shifts across linguistic levels"; and instruction-tuning encourages attention heads in lower layers to learn word-word relations with instruction verbs.

**Strengths:**

I will describe strengths, weaknesses, and questions together in the format below.


# Overall

Overall, the paper feels somewhat exploratory and still unrefined in its methods, and its takeaways or intended impact. The authors propose a general set of methods for LLM interpretability -- which seem useful and applicable beyond instruction-tuning, as the authors observe -- but, as a consequence, the application to instruction-tuning feels somewhat less motivated, and the findings don't lead to any clear insights about future work, how instruction-tuning works, or how to improve the instruction-tuning process. I also identify a few gaps in the authors' assessment of the results, which are somewhat qualitative in certian places. I feel that the paper could be improved substantially, but lays a good methodological foundation and have the potential to "shine more light" on a process that isn't particularly well understood yet. However, I feel that revisions, described below, would make the paper much stronger.

# Major Comments

* A major concern I have with this paper is that it isn't clear what it ultimately delivers to the reader: the results don't seem particularly surprising, and also don't lead to clear directions for improving models, training, or instruction-tuning. It seems obvious that instruction-tuning words are critical for models to understand, and that instruction-tuned models would attend strongly to them -- they are the semantic crux of the inputs. This would have been predictable, and indeed strongly expected, without the methods. What additional insight do we learn here, and how can we use this scientific knowledge to improve models? I am not sure after reading the paper.

* In a few places, the quantitative evaluation is somewhat informal and could be improved.
  - Section 3.1 is a particularly bad example of this: it uses a visualization of (if I am reading it correctly) a single prompt-response element in LLaMA and Vicuna to make its claim. This is achieved by a heatmap which requires 5 different annotation boxes to display the trends to the reader. A more quantitative analysis of these results would be more comprehensive and persuasive. As it is, the analysis is somewhat unclear (what are the "certain context lines that appear less bright"?) and hard to make sense of (is this a big effect? how does it compare to other examples, or non-instruction-tuning inputs?).
  - Similarly, many of the tables could improve the results presentations. For example, the null hypothesis being tested with the reported p-values is never reported (I assume it is that the difference between the two values is zero, but I am not sure). But, there are many details I am unsure of in these tables: what are the "instruction words" in Table 1, and how are they identified? Why are 5 of the top 69 principal components selected for Table 2, and where are the rest?

* In places it is difficult to follow the paper, due to a mix of missing context (I would suggest to move the related work to Section 2, since a great deal of the investigation here seems to be motivated by phenomena observed in prior work; the datasets are also never clearly introduced and explained, familiarity with them is assumed), terms that are not defined (I still don't fully understand the concept of a "linguistic level" and didn't find its definition in the paper), and typos (see below). The authors also sometimes skip over what seem to be important claims about their methods and metrics ("It's crucial to recognize that a word with a lower coenfidence doesn't necessarily imply it is a trivial word"; "if we simply map...it might give us the full picture of all these patterns"). I would suggest extensive revisions to clearly explain these details in terms of the provided equations (and would appreciate some clarification on those two sepcific phrases in the response).

* Since the authors present a general "toolkit" for interpretability in LLMs, it seems appropriate to consider applying this analysis to other contexts, or at least to clearly explain (1) where else these tools could be applied, and (2) how another user could utilize the authors' code (I would like a clearer explanation than "we will release our code and data upon acceptance" to understand how accessible and usable these tools would really be to the community).

# Minor Comments

* Please provide citations for the "well-accepted belief" regarding instruction-tuning described in Section 1.

* Based on the discussion in Section 1, it seems that Finding 3 is almost the same bottom-line finding as 1: "pre-trained models become adept at recognizing instruction words during instruction fine-tuning" (Finding 3, last sentence of bulleted paragraph in Section 1) could easily also describe Finding 1.


* The authors switch between "instruction tuninng" and "instruction fine-tuning" in the paper -- are these describing the same method? If so, please pick one and stick to it (instruction tuning seems more conventional).

* Please provide more details on the labeling of responses in Sectoin 3.2.2. Who labels them, and using what criteria?

* Please describe how the "new vocabulary derived from ShareGPT" is created in 4.2.

* Please provide more details regarding the PCA in Table 2: for example, seeing the cumulative proportion of variance explained would be useful in assessing whether the 69th-ranked vector is a meaningful one to be interpreting.


# Typos etc.

There are many typos in the paper; I list a few here but it needs a careful revision.
* P1: "what the encoded knowledge evolves" -> how the encoded knowledge evolves
* P2: "LLMs that billions of parameters."
* P2: "as their transition from language modeling to instruction following." -> as they transition
* P2: "our analyis to these concepts"
* P3: "a N-length" --> an N-length
* Paper switches between LLaMa, LLaMA, and Llama
* P4: "would receives more"
* P6: "we crease" --> we create
* P6: "the well interpretability of"

**Weaknesses:**

see above

**Questions:**

see above

---

> ### Author Response · Authors · 2023-11-15
> **Response to Reviewer e43k - Part 1/3**
>
> Thanks for your comments, we address the concerns you raised as follows.
>
> **Major-1: It isn't clear what it ultimately delivers to the reader.**
>
> This paper contributes novel insights into *how* instruction-tuned models generate responses aligned with human preferences. This is not fully explored in prior research. While the importance of instruction words seems intuitive, our study empirically reveals how self-attention and feed-forward networks distinctively contribute to this capability.
> Specifically, the self-attention heads from the bottom layers are responsible for recognizing the instructions by encoding more word-word relations with instruction verbs, while feed-forward networks rotate their pre-trained knowledge toward user preferences.
>
>
> Our research provides specific recommendations for building models with instruction-following ability. For example, our results suggest that training self-attention heads is more crucial, particularly those in the lower layers, when full-parameter fine-tuning is not feasible due to limited resources. Additionally, for tasks that need a substantial amount of pre-trained knowledge, such as open-domain QA, it's advantageous to train feed-forward networks concurrently. These guidelines are grounded in the findings of empirical studies [1, 2, 3].
>
> Furthermore, our study observed some unexpected phenomena. Contrary to conventional understanding, language models encode a big proportion of semantic knowledge in the bottom layers, while the top layers encode considerably less. Additionally, our findings suggest that models may rely on inherent language modeling abilities, rather than instruction-following abilities, to complete user tasks. We also identified the 'lost-in-the-middle' issue from the pre-trained models, where models focus on the beginning and end of input texts, overlooking the middle sections. These findings are crucial for refining model architectures and training processes.
>
> Moreover, our study introduces the concept of 'importance density' as a new metric for evaluating how models utilize input prompts, with potential applications in model training (e.g., the reward signals for RLHF), backdoor detections, hallucination detections, and so on.
>
> > [1] Hu, Edward, et al. LoRA: Low-Rank Adaptation of Large Language Models. ICLR. 2022.
> >
> > [2] Juletx. Alpaca-LoRA. https://github.com/tloen/alpaca-lora. 2023.
> >
> > [3] Sun, Xianghui, et al. A Comparative Study between Full-Parameter and LoRA-based Fine-Tuning on Chinese Instruction Data for Instruction Following Large Language Model. arXiv. 2023.
>
> __Major-2.1: Section 3.1 makes a claim by analyzing the visualization of a single instance.__
>
> We design the experiments in Section 3.1 as a qualitative analysis. Although we discuss our observation with one single instance in the main paper, we confirm it holds for more cases and provide their saliency maps in Appendix A.2. We quantify this observation in Section 3.2. Thus, both visualization (Section 3.1) and quantitative experiments (Section 3.2) demonstrate that instruction words constantly guide the generation process as the key behavior of LLMs to generate responses aligned with humans. The revised version will emphasize the connection between results in Section 3.1 and Section 3.2.
>
> __Major-2.2: Improving the result presentations of tables.__
>
> Thanks for the suggestion, we will take all of them to our revised version. Here are direct responses to your concerns:
>
> (1) Yes, the null hypothesis is the difference between the two values is zero. We want to show that the difference between the experiment and control groups is statistically significant.
>
> (2) The instruction part of an input prompt includes sentences that describe background information (optional) and the required actions for a task. For example, giving the user input prompt "fix grammar: me and him goes to an stores for buy a apples.", the instruction part would be "fix grammar:". The authors manually annotate the instruction parts of 632 samples from the three datasets. Please see Appendix B.1 for more details.
>
> (3) Table 2 only presents 5 interpretation cases out of 69 because we cannot include the cases that are hard to present in the main paper with such limited spaces. In Appendix C.2 and Figure 7, we demonstrate that over 80% of principal components from the last layer of Vicuna/LLaMA could be interpreted by ChatGPT. In addition, Table 9 and Table 10 include more interpretable principal components. Overall, the word lists always show clear and concise concepts to humans and we will release the full annotated dataset once the paper is accepted.

---

> > ### Comment · Reviewer_e43k · 2023-11-21
> >
> > Thanks to the authors for their response and revisions.
> >
> > I feel that my original concerns with the paper still haven't been fully addressed; in particular, it is not entirely clear to me how the authors' purported insights into the dynamics of the fine-tuning process can lead to improvements in fine-tuning, architecture, training, etc.
> >
> > I acknowledge that the authors have made extensive revisions to the paper -- however, now the paper is more than a page over the limit (page limit is 9 pages https://iclr.cc/Conferences/2024/AuthorGuide ; the current response is slightly less than 10.5 pages!). As a result, it is hard to review the paper without understanding what its contents would be when fit within the 9 page limit. So, I opt to leave my score unchanged.

---

> > > ### Author Response · Authors · 2023-11-21
> > > **Response to additional feedback from Reviewer e43k with Revision-V2 submitted**
> > >
> > > Thank you for your continued feedback and engagement with our paper.
> > >
> > > We understand your concerns regarding the direct applicability of our insights into the fine-tuning process. To clarify, our research aims to offer a comprehensive framework that synthesizes various empirical phenomena observed in recent studies. While we may not provide immediate, direct improvements in fine-tuning, architecture, or training, our work lays the groundwork for future research in these areas. The theoretical perspectives and toolkits developed in this study are designed to inform and inspire subsequent practical advancements.
> > >
> > > Regarding the paper's length, we appreciate your pointing out the discrepancy. We have diligently worked to condense our manuscript to adhere strictly to the 9-page limit set by the conference. In this revised version (V2), we have streamlined our content while ensuring that the core contributions and insights remain clear and accessible. The end of the Introduction and Related Work sections now more explicitly highlight the significance and potential impact of our work in the field.
> > >
> > > We hope that these revisions more adequately address your concerns and look forward to your feedback.

---

> ### Author Response · Authors · 2023-11-15
> **Response to Reviewer e43k - Part 2/3**
>
> __Major-3: In places, it is difficult to follow the paper due to missing context.__
>
> Thanks for the suggestion, we will include more details in our revised paper, including but not limited to the following:
>
> (1) We will discuss more observed phenomena in prior work that motivate this research in Section 2.
>
> (2) We follow the discipline classification system in linguistics to discuss the "Linguistic levels". As mentioned in Section 4.2, the linguistic levels consider four levels of knowledge: phonology, morphology, syntax, and semantics. For more discussions of these linguistic disciplines, we would like to refer to the reference [1].
>
> (3) We make the statement that "It's crucial to recognize that a word with a lower confidence doesn't necessarily imply it is a trivial word." in Section 3.1.1 with the following considerations. Formally, the next-word prediction process could be written as $p(y|x)$ where $x$ is the previous context and $y$ is a word. With Bayes equation, we have $p(y|x)\propto  p(x|y)\cdot p(y)$. Regardless of the context $x$, the trivial words have a higher prior probability since they are much more common in the general corpus. Regarding the context $x$, $p(x|y)$is smaller for the $y$ are non-trivial words unless $(x,y)$ shows a common phase. Overall, a model is more confident if it tries to generate a trivial word, while it is less confident when it tries to predict a semantic word.
>
> (4) We make the statement that "If we simply map ... it might not give us the full picture of all these patterns" in Section 4.1 with the following considerations. This Section aims to interpret the knowledge encoded in feedforward networks, and the most straightforward way is interpreting neurons (vectors of the weight matrix). However, many studies have pointed out that neurons are polysemantic [2, 3, 4, 5], leading to poor interpretability for humans. Therefore, we propose to decompose these neurons (patterns) via principal component analysis.
>
> > [1] James J Thomas. Illuminating the path:[the research and development agenda for visual analytics]. IEEE Computer Society, 2005.
> >
> > [2] Trenton Bricken, et. al. Towards Monosemanticity: Decomposing Language Models With Dictionary Learning. _Anthropic Blog_, 2023.
> >
> >  [3] Arora, Sanjeev, et al. "Linear algebraic structure of word senses, with applications to polysemy." *Transactions of the Association for Computational Linguistics* 6 (2018): 483-495.
> >
> >  [4] Elhage, Nelson, et al. "Toy models of superposition." *arXiv preprint arXiv:2209.10652* (2022).
> >
> >  [5] Cunningham, Hoagy, et al. "Sparse autoencoders find highly interpretable features in language models." *arXiv preprint arXiv:2309.08600* (2023).
>
> **Major-4: How will the developed XAI toolkit benefit future works?**
>
>
> We are planning to release the toolkit as a Python package (it could be pip installed). The toolkit will support various language models implemented by the Huggingface Transformers library well. The package will initially include these three proposed methods, but later, it will include other explanation methods.
>
> The proposed three methods could be applied to various tasks and challenges. For example, the importance density could not only be used to evaluate the instruction following ability, but also other scenarios. Retrieval augmented generation (RAG) is one possible scenario, where they attach the retrieved documents as extra knowledge to the language models by assuming the models always utilize the information from the prompt. To verify this assumption during inference, we could compute the importance density over the retrieved documents.
>
> __Minor-2:  It seems that Finding 3 is almost the same bottom-line finding as 1.__
>
> The main difference between Finding 1 and Finding 3 is in their perspective. Finding 1 is made based on the outcomes from the gradient-based attribution method over prompt-response pairs, while Finding 3 is drawn from a global explanation over model weights. Therefore, Finding 1 provides evidence that LLMs align with human preference by identifying instruction words, while Finding 3 is a deeper discussion of this phenomenon by indicating that LLMs acquire this ability by encoding more word-word patterns with instruction verbs.
>
> __Minor-3: Instruction-tuning or Instruction fine-tuning.__
>
> Yes, they are the same thing in this paper. The revised version will take the terms constantly.
>
> __Minor-4: Labelling details of Section 3.2.2.__
>
> The authors manually labeled a total of 632 prompt-response pairs for this experiment. Specifically, we consider the helpfulness of the response as the ability of instruction following. Therefore, if a response is helpful to the user, then we label it with “followed”. Please check Appendix B.1 for more details.

---

> ### Author Response · Authors · 2023-11-15
> **Response to Reviewer e43k - Part 3/3**
>
> __Minor-5: Constructing new vocabulary with the ShareGPT dataset.__
>
> The tokenizing pipeline for user prompts from the ShareGPT dataset remains the following steps, including tokenizing with ``nltk.word_tokenize``, part-of-speech tagging with  ``nltk.pos_tag``, and lemmatizing with ``nltk.wordnet.WordNetLemmatizer``.
>
>
> __Minor-6: More results regarding to PCA in Table2.__
>
> Generally, the variance explained drops smoothly. For the last layer of Vicuna, the first 300 principal components totally explain 22.28% of the total explained variances, where 300 is the number of components we studied in Section 4. The revised version will include the plot of the explained principal components.
>
> __Typos.__
>
> Thanks for pointing out all these typos. We will address them in our revised version. We will also make another detailed check on the whole paper.

---

> > ### Comment · Reviewer_e43k · 2023-11-18
> > **Have revisions been made to the paper?**
> >
> > Before I respond to the authors' follow-up, I would like to know: has the paper actually been revised to reflect these concerns? If so, please clarify where the changes have been made, as I can't locate any changes in the current PDF. The author-reviewer discussion period allows authors to revise their submission.

---

> > > ### Author Response · Authors · 2023-11-20
> > > **Revision-V1 has been submitted**
> > >
> > > Dear Reviewer e42K,
> > >
> > > We have submitted our first revision to the system, feel free to make any further comments or suggestions.
> > >
> > > Thank you!
> > > Authors of Submission 2980

---

### Author Response · Authors · 2023-11-20
**Revision-V1 of paper 2980 has been submitted.**

Dear Reviewers,

We are pleased to submit the first revision of our paper, "From Language Modeling to Instruction Following: Understanding the Behavior Shift in LLMs after Instruction Tuning." We have marked all changes in blue for easy identification. Below is a summary of the key revisions:

__1. Improvement in Sections 3.1 and 3.2:__ Section 3.1 now focuses on a _qualitative analysis_ of instruction words, positing that they constantly guide the response generations to align with human intentions. This hypothesis is _quantitatively verified_ in Section 3.2 through experiments with importance density scores.

__2. Improved Writing Clarity:__ (1) Highlighted the key contribution of this research. (2) Relocated the related work to Section 2, setting a more coherent background for our research (e.g., "lost-in-the-middle''). (3) Clarified terms, such as "linguistic levels" and "instruction words," and expanded discussions for better understanding. (4) Clarified statements, such as "if we simply map...it might give us the full picture of all these patterns", "It's crucial to recognize...it is a trivial word", "the derivation of S_{n,m} from I_{n,m} and its intuition". (5) Corrected typos and miscellaneous errors.

__3. Refined Presentation of Experimental Results:__ (1) Added detailed experimental settings in the main text, with comprehensive settings in the appendix. (2) Introduced a brief overview of the results in tables and figures before delving into in-depth analysis. (3) Made the null hypothesis for each experiment that useed the student T-test for validation.

__4. Additional Experimental Results:__ (1) Appendix B.1.1 now includes a comparative study of importance density in responses generated by Vicuna and LLaMA for comparison. (2) Appendix E.1 presents findings on accumulated explained variance using PCA over feed-forward network weights.

__5. Further Theoretical Justification:__ Appendix A now contains a theoretical proof of the linear approximation of the importance score using the first-order Taylor extension.

We look forward to any further feedback or suggestions during the rebuttal period and are committed to improving our work based on your valuable insights.

Thank you for your consideration.

Sincerely,

Authors of Submission 2980